# An inter-species protein–protein interaction network across vast evolutionary distance

Quan Zhong[1,2,3,†,****], Samuel J Pevzner[1,2,4,5,†], Tong Hao[1,2], Yang Wang[1,2], Roberto Mosca[6], Jörg Menche[1,7], Mikko Taipale[8], Murat Taşan[1,9,10,11], Changyu Fan[1,2], Xinping Yang[1,2], Patrick Haley[1,2], Ryan R Murray[1,2], Flora Mer[1,2], Fana Gebreab[1,2], Stanley Tam[1,2], Andrew MacWilliams[1,2], Amélie Dricot[1,2], Patrick Reichert[1,2], Balaji Santhanam[1,2], Lila Ghamsari[1,2], Michael A Calderwood[1,2], Thomas Rolland[1,2], Benoit Charloteaux[1,2], Susan Lindquist[8,12,13], Albert-László Barabási[1,7,14], David E Hill[1,2], Patrick Aloy[6,15], Michael E Cusick[1,2], Yu Xia[1,16,***], Frederick P Roth[1,9,10,11,17,**] & Marc Vidal[1,2,*]

## Abstract

In cellular systems, biophysical interactions between macromolecules underlie a complex web of functional interactions. How biophysical and functional networks are coordinated, whether all biophysical interactions correspond to functional interactions, and how such biophysical-versus-functional network coordination is shaped by evolutionary forces are all largely unanswered questions. Here, we investigate these questions using an "inter-interactome" approach. We systematically probed the yeast and human proteomes for interactions between proteins from these two species and functionally characterized the resulting inter-interactome network. After a billion years of evolutionary divergence, the yeast and human proteomes are still capable of forming a biophysical network with properties that resemble those of intra-species networks. Although substantially reduced relative to intra-species networks, the levels of functional overlap in the yeast–human inter-interactome network uncover significant remnants of co-functionality widely preserved in the two proteomes beyond human–yeast homologs. Our data support evolutionary selection against biophysical interactions between proteins with little or no co-functionality. Such non-functional interactions, however, represent a reservoir from which nascent functional interactions may arise.

**Keywords** Cross-species complementation; Network evolution; Selection
**Subject Categories** Evolution; Genome-Scale & Integrative Biology; Network Biology
**Mol Syst Biol. (2016) 12: 865**

## Introduction

Complex interactome networks of interacting genes and gene products underlie most genotype–phenotype relationships. Hundreds of thousands of molecular interactions are coordinated at the scale of

1 Center for Cancer Systems Biology (CCSB) and Department of Cancer Biology, Dana-Farber Cancer Institute, Boston, MA, USA
2 Department of Genetics, Harvard Medical School, Boston, MA, USA
3 Department of Biological Sciences, Wright State University, Dayton, OH, USA
4 Department of Biomedical Engineering, Boston University, Boston, MA, USA
5 Boston University School of Medicine, Boston, MA, USA
6 Joint IRB-BSC-CRG Program in Computational Biology, Institute for Research in Biomedicine (IRB Barcelona), The Barcelona Institute of Science and Technology, Barcelona, Catalonia, Spain
7 Center for Complex Network Research (CCNR) and Department of Physics, Northeastern University, Boston, MA, USA
8 Whitehead Institute for Biomedical Research, Cambridge, MA, USA
9 Departments of Molecular Genetics and Computer Science, University of Toronto, Toronto, ON, Canada
10 Donnelly Centre, University of Toronto, Toronto, ON, Canada
11 Lunenfeld-Tanenbaum Research Institute, Mt. Sinai Hospital, Toronto, ON, Canada
12 Department of Biology, Massachusetts Institute of Technology, Cambridge, MA, USA
13 Howard Hughes Medical Institute, Massachusetts Institute of Technology, Cambridge, MA, USA
14 Department of Medicine, Brigham and Women's Hospital, Harvard Medical School, Boston, MA, USA
15 Institució Catalana de Recerca i Estudis Avançats (ICREA), Barcelona, Spain
16 Department of Bioengineering, McGill University, Montreal, QC, Canada
17 Canadian Institute for Advanced Research, Toronto, ON, Canada
 *Corresponding author. Tel: +1 617 632 5180; E-mail: marc_vidal@dfci.harvard.edu
 **Corresponding author. Tel: +1 416 946 5130; E-mail: fritz.roth@utoronto.ca
 ***Corresponding author. Tel: +1 514 398 5026; E-mail: brandon.xia@mcgill.ca
 ****Corresponding author. Tel: +1 937 775 3571; E-mail: quan.zhong@wright.edu
 †These authors contributed equally to this work

the genome, transcriptome, and proteome, forming functional networks. Reflecting this coordination, physically interacting proteins are "co-functional", *that is,* share highly related molecular functions and/or similar expression or phenotypic profiles (Ge *et al*, 2001; Gunsalus *et al*, 2005; Yu *et al*, 2008; Vidal *et al*, 2011), at a rate that is one to two orders of magnitude higher than random pairs. It is increasingly clear that biological networks are heterogeneous, containing interactions that could be functionally insignificant (Levy *et al*, 2009; Venkatesan *et al*, 2009; Rolland *et al*, 2014) or have non-adaptive evolutionary origins (Fernandez & Lynch, 2011; Sorrells & Johnson, 2015). The extent to which biophysical interactions may occur in the absence of adaptive evolution and how such interactions distribute in biological networks remain poorly understood. This is largely due to the difficulty in identifying biophysical interactions that are fully uncoupled from biological functions.

Here, we used an "inter-interactomic" approach to test the intrinsic ability of proteins to interact apart from any direct selective pressure. We investigated (i) to what extent two evolutionarily distant proteomes are capable of forming a biophysical inter-species "inter-interactome" network and (ii) how such an inter-interactome might correlate with intra-species functional relations (Fig 1A). As a proof of principle, we tested for interactions between the proteomes of two species—yeast (*Saccharomyces cerevisiae*) and human—separated by approximately one billion years of evolution.

Previous studies on interactome evolution have focused on "rewiring" of interactions among pairs of proteins that are phylogenetically related within or between species (paralogs and orthologs, respectively) (Walhout *et al*, 2000; Matthews *et al*, 2001; Wagner, 2001; Yu *et al*, 2004; Sharan & Ideker, 2006; Grove *et al*, 2009; *Arabidopsis* Interactome Mapping Consortium, 2011; Das *et al*, 2013; Reece-Hoyes *et al*, 2013; Reinke *et al*, 2013). The approach taken here, experimental inter-interactome mapping, encompasses large proportions of proteins without readily detectable homologs in the opposing species, probing evolutionary possibilities beyond phylogenetic conservation (Fig 1B). Loss of ancestral binding properties in yeast and human proteins or gain of adventitious binding in the absence of selection against deleterious interactions (Zarrinpar *et al*, 2003) may lead to an inter-interactome that differs starkly from intra-species networks and has little correspondence to intra-species functional relationships.

# Results

## Inter-species interactions of human proteins with conserved functions in yeast

The phenotypes conferred by mutations in a particular gene can often, but not always, be "rescued" by heterologous expression of orthologs from distantly related species (Kachroo *et al*, 2015). We first determined to what extent human proteins that can functionally rescue yeast mutations ("rescuers") retain mutual human–yeast inter-species interactors with the yeast orthologs they are capable of rescuing ("rescuees") (Fig 2A). Using the yeast two-hybrid (Y2H) system (Dreze *et al*, 2010), we screened 172 human rescuers (Appendix Supplementary Methods) as Gal4 activation domain

(AD) hybrid proteins (AD-X$_{human}$) against approximately two-thirds of all yeast proteins expressed as DB domain hybrids (DB-Y$_{yeast}$). We compared the obtained set of inter-species biophysical interactions to a high-quality set of literature-curated intra-species interactions involving the corresponding yeast rescuees (Table EV1). Among 46 human–yeast inter-species interactions identified, ~25% involve an interactor that is shared between rescuers and rescues (10-fold more than would be expected by chance: empirical *P*-value = 0.001, Fig 2A, bottom). Of the eight pairs of rescuers and rescues recovered as sharing yeast interactor(s), seven (88%) have functions similar to their mutual interactors (Fig 2A). For example, human MLH1 and yeast Mlh1, and their mutual interactor Ntg2, are all involved in DNA repair (Fig 2A). Thus, interactions between two evolutionarily distant proteomes can derive from "ancestral" interactions, which likely took place in their last common ancestor. Of the eleven yeast proteins recovered as mutual interactors between rescuers and rescues, three (27%) had no homolog in human (Fig 2A), consistent with the idea that ancestral protein-binding sites evolve new interactions with non-phylogenetically conserved proteins.

## Systematic mapping of human–yeast inter-species protein–protein interactions

To systematically determine the extent to which biophysical interactions occur between evolutionarily distant proteomes, we generated a high-quality proteome-wide inter-interactome network map of human–yeast binary inter-species interactions. To allow us to compare the properties of the resulting inter-interactome network to those of the two parent intra-species networks, we designed an inter-species search space (Fig 2B) that matches the sets of proteins previously used to generate intra-species interactome maps for human (Rual *et al*, 2005) and yeast (Yu *et al*, 2008). We identified inter-species interactions by a single-pass Y2H screen followed by quadruplicate pairwise tests and orthogonal validation using the modified luminescence-based mammalian interactome ("LUMIER") assay (Taipale *et al*, 2012). In all, we performed Y2H screens between 7,240 human (AD-X$_{human}$) (Rual *et al*, 2005) and 3,778 yeast (DB-Y$_{yeast}$) (Yu *et al*, 2008) proteins, corresponding to ~28 million yeast–human protein pairs. We identified 1,583 inter-species interactions between 566 yeast and 471 human proteins, including 284 pairs for which neither protein was conserved between the two species (Table EV2). Interspecies pairs had a validation rate comparable to that of a positive reference set of well-documented human intra-species protein–protein interactions (Venkatesan *et al*, 2009) (Fig 2C). Assuming that the detectability of interactions (Venkatesan *et al*, 2009) is similar between intra- and inter-species Y2H screens, and taking into account the size of our search space relative to the full inter-species space, but leaving aside the complexity of human alternatively spliced isoforms, our yeast–human inter-species interactome (YHII-1) network suggests that the yeast and human proteomes could mediate $10^4$–$10^5$ biophysical inter-species interactions (Fig 2D). Thus, intra- and inter-species networks appear to have similar size, ruling out models where the number of inter-species interactions would either be extremely small due to the lack of selection for functional interactions, or extremely large due to the lack of selection against deleterious interactions.

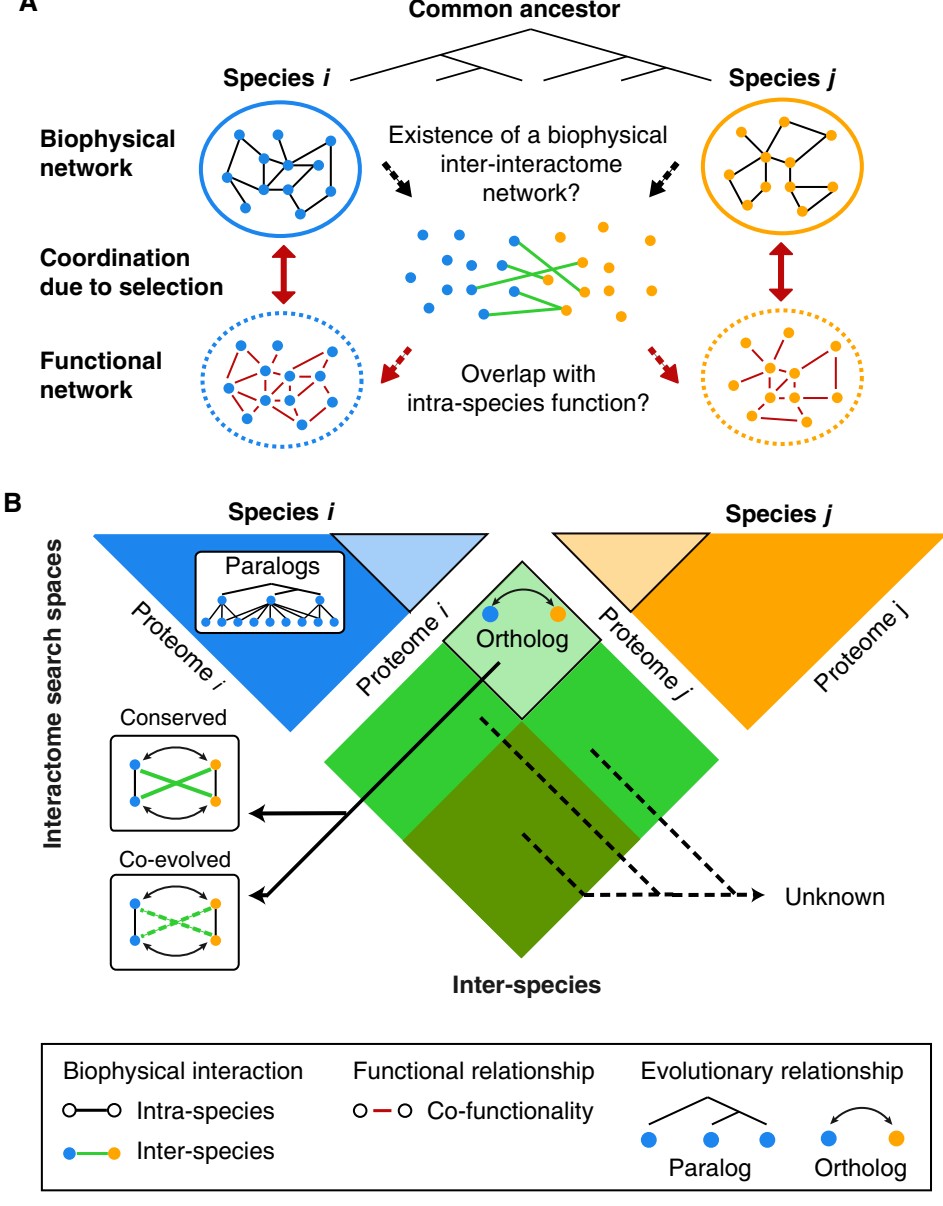

**Figure 1.  The concept of systematic "Inter-interactome" mapping at the proteome scale.**

A   Using inter-species inter-interactome networks to investigate how evolution shapes the coordination between biophysical and functional networks.

B   Proteome-scale inter-interactome mapping encompasses large numbers of proteins that are not conserved between the two species and their propensity to form inter-species interactions unknown.

**Ancestral origins of inter-species protein–protein interactions**

We evaluated the extent to which inter-species interactions may originate from evolutionarily conserved protein-binding mechanisms. First, where homology relationships were present, YHII-1 interactions were 15–20 times more likely than expected by chance to overlap with interactions from the two parent systematic intra-species maps (Rual *et al*, 2005; Yu *et al*, 2008) (Fig 3A). Such inter-species interactions likely involve conserved binding properties retained in human–yeast homologs. Pairs of homologs were found to interact with proteins that are not conserved between human

and yeast (Table EV3), consistent with ancestral protein-binding sites evolving new interactions with non-phylogenetically conserved proteins. Second, we used three-dimensional structural evidence available for intra-species interactions to find that a small but significant number of inter-species interactions correspond to conserved protein-binding sites (Appendix Supplementary Methods and Table EV4). Third, considering proteins that have known interaction domains, nearly 25% of inter-species interactions of such proteins can be explained by domain–domain interactions (Table EV5). This is significantly higher than expected by chance (Fig 3B, empirical *P*-value = 0.001). Fourth, interaction profile

 

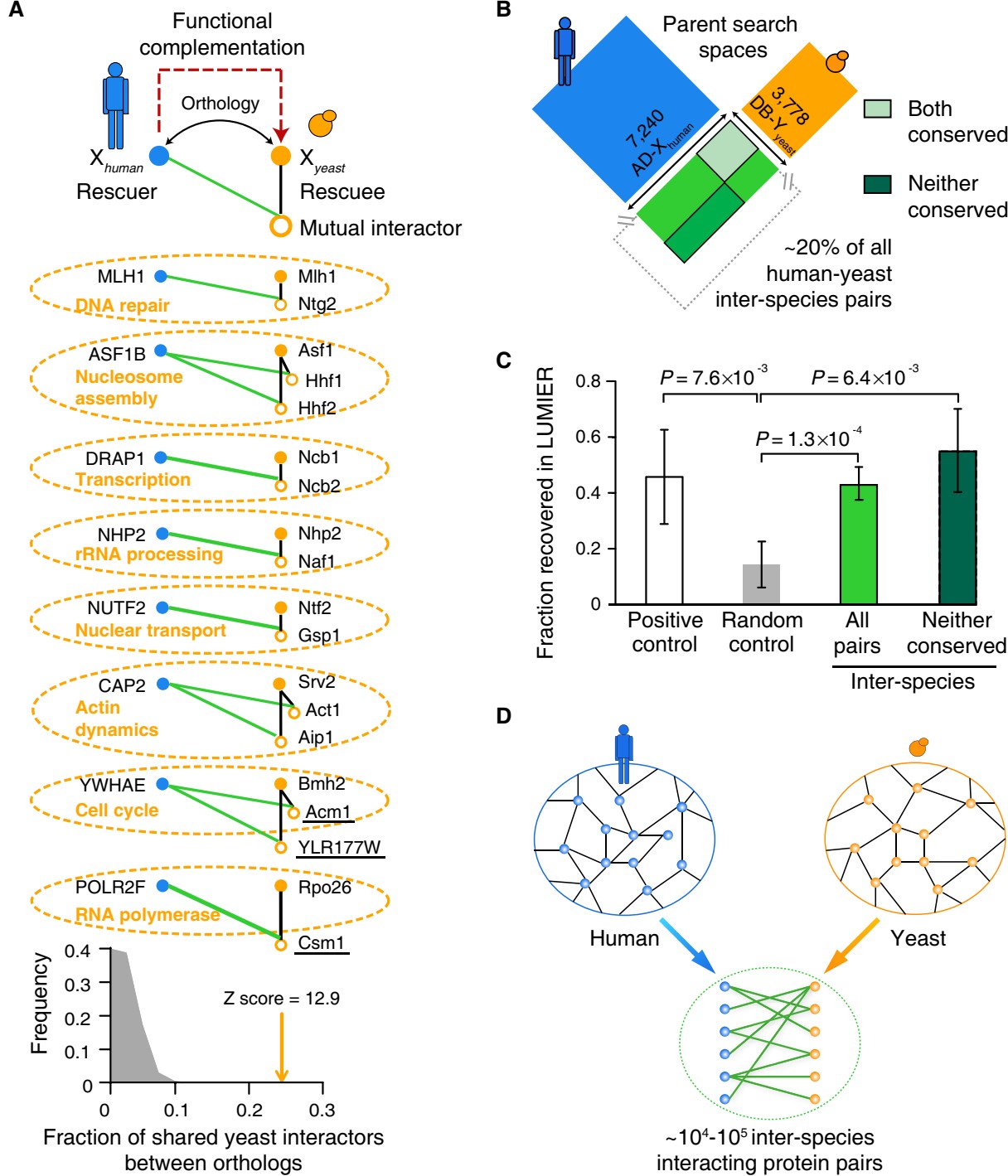

**Figure 2.  A proteome-scale human–yeast inter-interactome.**

A   Inter-species interactions of human rescuers reveal eleven yeast interactors shared by yeast rescuees. Fraction of interactors shared between yeast and human orthologs relative to randomized controls, empirical *P*-value = 0.001. Dotted ellipses mark proteins with shared function. Three proteins not conserved from yeast to human are underlined.

B   Search space of the systematic inter-species network map matching two parent intra-species networks.

C   Fractions of pairs recovered by the orthogonal LUMIER validation assay for: the positive control, the human positive reference set (PRS) (Venkatesan *et al*, 2009); the random control, the yeast–human random reference set (RRS) (Venkatesan *et al*, 2009); and inter-interactome (YHII-1) sample pairs in the full search space as well as in the subspace containing only proteins not conserved between human and yeast. Error bars: standard error of the proportion. *P*-values determined using chi-square test with Yates correction.

D   Projected number of inter-species interactions between the human and yeast proteomes considering the coverage of the YHII-1 search space and the standard sampling and assay sensitivity using the same Y2H pipeline (Venkatesan *et al*, 2009).

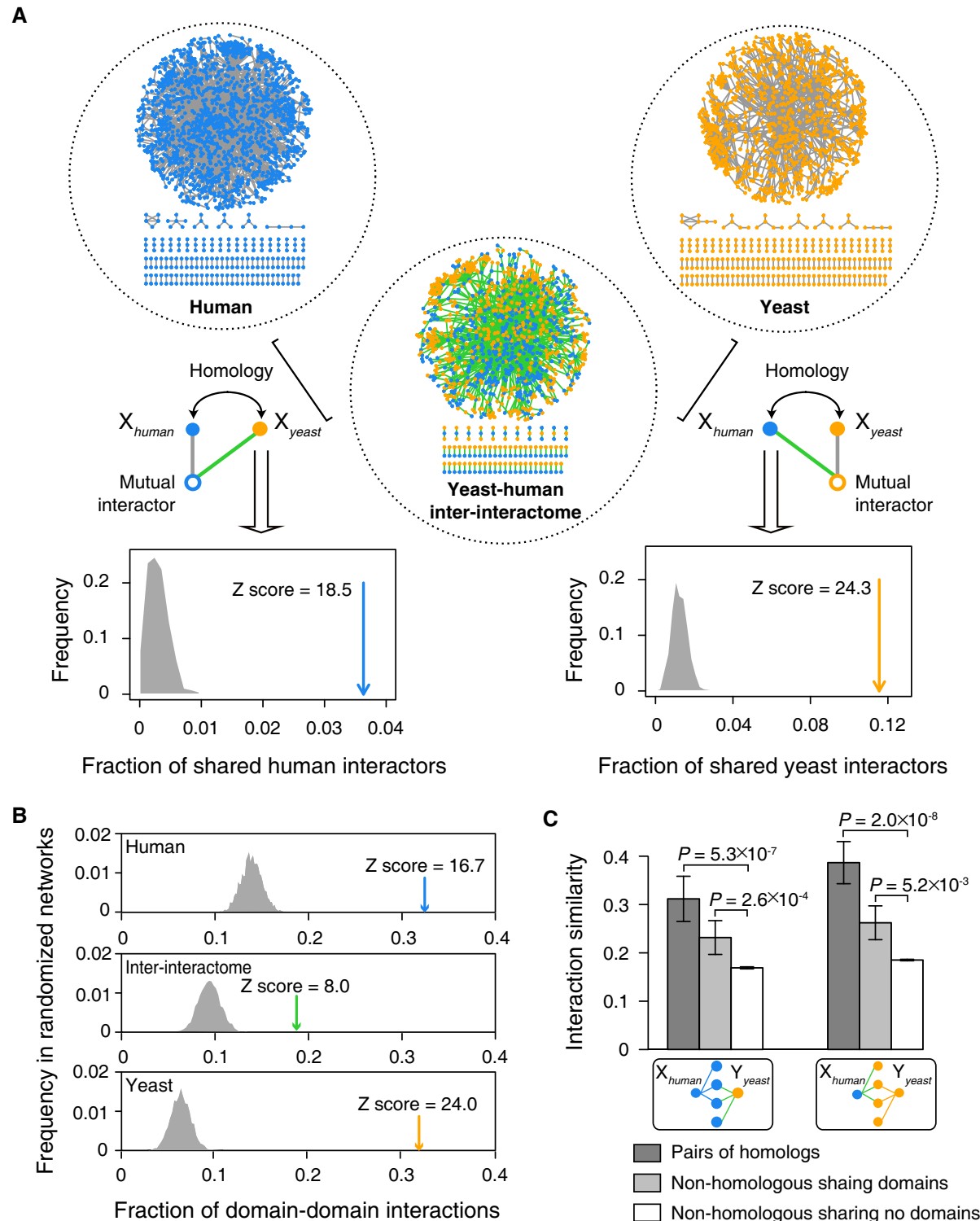

**Figure 3.  Conserved protein binding underlying inter-species interactions.**

A  Fractions of shared interactors between human and yeast homolog pairs (arrows) relative to randomized controls (gray-shaded areas), using intra-species human HI-1 (top left) or yeast YI-1 (top right) as the reference network, empirical *P*-values = 0.001. Blue nodes: human proteins; orange nodes: yeast proteins; gray edges: intra-species yeast–yeast or human–human interactions; green edges: yeast–human inter-species interactions.

B  Fractions of interactions plausible through high confidence domain–domain interaction pairs (Yellaboina *et al*, 2011) in the human (HI-1, blue arrow), inter-interactome (YHII-1, green arrow), and yeast (YI-1, orange arrow) networks relative to randomized network controls (gray-shaded areas), empirical *P*-value = 0.001.

C  Interaction profile similarity of human and yeast protein pairs sharing at least one common human (left) or yeast (right) interactor. *P*-values determined by Mann–Whitney *U*-test.

similarity indexes between human and yeast homologs measured using inter-species interactions were significantly higher than other protein pairs that share at least one common interactors (Fig 3C). Consistent with conserved protein interaction domains underlying inter-species interactions, significantly higher interaction profile similarity was also observed for human–yeast protein pairs that are not human–yeast homologs but have at least one predicted protein domain in common (Fig 3C). Altogether, a significant proportion of inter-species interactions derive from ancestral interactions, notwithstanding the possible existence of non-ancestral, or adventitious, biophysical interactions between the yeast and human proteomes.

Essential yeast proteins appear to have more inter-species interactors than non-essential proteins (mean number of interactors for yeast proteins: essential, 3.3; non-essential: 2.6; *P*-value = 0.009 by Mann–Whitney *U*-test). Essential yeast proteins are also enriched among proteins that form inter-species interactions overlapping with intra-species interactions (odds ratio 1.8, *P*-value = 0.001 by Fisher's exact test). Consistent with inter-species interactions corresponding to evolutionarily conserved gene functions, we found a significant enrichment of human proteins in the inter-interactome that can complement essential functions of their yeast homologs (Kachroo *et al*, 2015) (Odds ratio 3.8, *P*-value is 0.03 by Fisher's exact test). These observations support our conclusions that inter-species interactions significantly correspond to ancestral protein-binding sites preserved in human and yeast proteomes. Conserved inter-species interactions likely underlie human–yeast cross-species functional complementation. Preservation of ancestral binding mechanisms may stem from evolutionary constraints on essential gene functions.

### Proteome-wide distribution of inter-species protein–protein interactions

To explore global patterns of human–yeast inter-species interactions across the two distantly related proteomes, we compared general interaction trends between YHII-1 and the two intra-species human (Rual *et al*, 2005) and yeast (Yu *et al*, 2008) parent networks. Co-evolution (Moyle *et al*, 1994), which modifies protein-binding interfaces while preserving interactions between conserved proteins, leads to loss of ancestral binding sites and incompatibilities between proteins and the orthologs of their interaction partners (Fig 1B). Such inter-species incompatibilities may underlie Dobzhansky–Muller interactions, originally hypothesized (Dobzhansky, 1936; Muller, 1942) and more recently verified (Presgraves *et al*, 2003; Brideau *et al*, 2006; Tang & Presgraves, 2009) to be a mechanism by which incompatible variants segregating within population drive speciation. Despite distant evolutionary separation, the density of the interactions found in the inter-species search space is comparable to that of the corresponding intra-species parent search spaces, even for pairs of proteins such that neither protein is conserved between human and yeast (Fig 4A). The same trend is true when we considered pairs of human–yeast proteins such that both are lineage-specific (*i.e.,* human and yeast proteins with only metazoan or fungal homologs respectively). Consistent with these findings, inter-species interactions are widespread and involve human and yeast proteins with little or no sequence conservation in the opposing proteomes (Fig 4B).

We next asked whether certain proteins might be more able to participate in biophysical interactions between diverged proteomes. We examined each protein domain with respect to its propensity to form inter-species versus intra-species interactions (Fig 4C). Three domains exhibit greater propensity for inter-species than intra-species interactions. Among them, the WD40 domain is well known to mediate protein interactions through recognition of diverse short peptides and linear motifs (Stirnimann *et al*, 2010). The zf-C3HC4 domain is a zinc finger subtype found primarily in ubiquitin–protein ligases that contributes to the specificity of their target selection (Deshaies & Joazeiro, 2009). Given that linear motifs can arise *de novo* more readily than complex and specific domain binding interfaces, such linear motifs might explain cases of adventitious binding in the inter-interactome. This hypothesis is consistent with the observation that the fraction of plausible domain–domain interactions in the inter-interactome is markedly lower than that in the yeast YI-1 or human HI-1 intra-species networks (Fig 3B). The reduced propensity of proteins containing WD40 or zf-C3HC4 domains to form interactions within species (Fig 4C) suggests selection against the evolution of non-functional linear motifs binding these domains in cellular networks, enhancing the binding and functional specificity of such motif-binding protein domains (Zarrinpar *et al*, 2003).

Intrinsically disordered regions of proteins are known for conferring conformational flexibility to binding partners (Dunker *et al*, 2005) and for a tendency to form promiscuous molecular interactions through mass-action effects (Vavouri *et al*, 2009). We found that human proteins with higher disorder content have greater propensities to form inter-species interactions (Pearson's correlation 0.17, *P*-value is 0.0002). Such a correlation is absent for intra-species interactions in the human HI-1 network (*P*-value is 0.2), consistent with disordered regions of proteins providing an increased tendency for adventitious inter-species interactions.

Together, these observations suggest that equivalent interaction densities of the inter-interactome and its intra-species parent networks are result of opposing evolutionary forces: frequent emergence of biophysical interactions uncoupled from pre-existing functional constraints versus evolutionary selection that preserves functional interactions and removes deleterious interactions in intra-species networks.

### Network properties of the human–yeast inter-interactome

Biological networks in diverse species across the kingdoms of life share local and global properties (Barabási *et al*, 2011; Vidal *et al*, 2011), which are thought to result from universal constraints on complex systems. Since inter-species interactions have not been subjected to direct selective pressures, the inter-interactome might exhibit different network features. Instead, the global topological properties of the inter-interactome are indistinguishable from those of the parental networks. All three networks exhibit similar scale-free, betweenness, disassortativity, and shortest path length properties (Fig 5A), and the degrees of yeast and human proteins in the inter-interactome correlate significantly with their degrees in the parent networks (Fig 5B).

In interactome maps, high-degree proteins, or "hubs", tend to be highly pleiotropic (Yu *et al*, 2008). The inter-interactome provides a

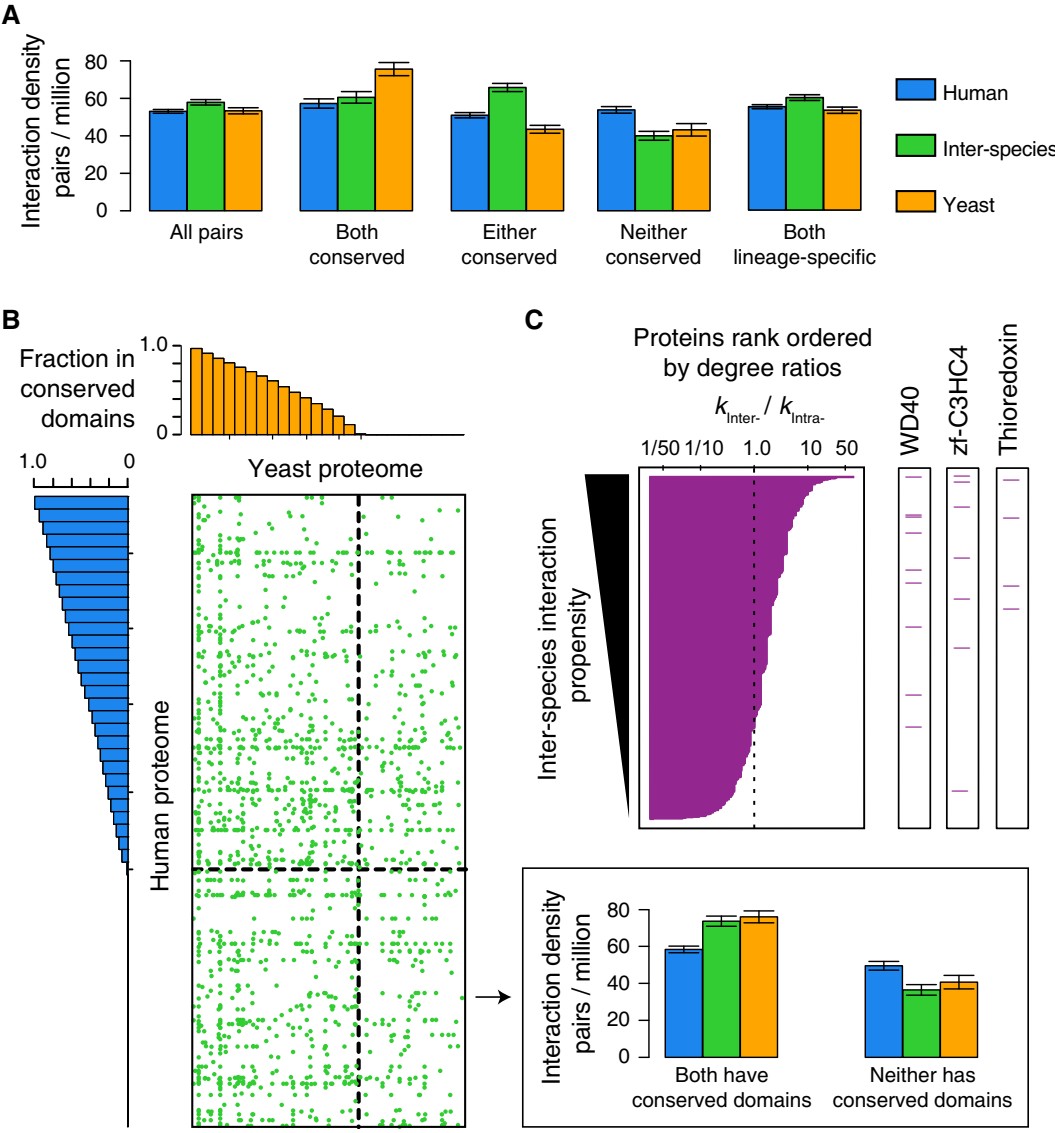

**Figure 4.  Proteome-wide distribution of inter-species interactions.**

A   Density of interactions within the full search spaces and the subspaces containing conserved or non-conserved human and yeast proteins. Error bars: standard error of the proportion.

B   Proteome-wide distribution of inter-species interactions (green dots). Human and yeast proteins are arranged and binned according to the fraction of their sequences found within protein domains present in both yeast and human proteomes. Histograms describe the minimum fractions of sequence in human (left) and yeast (top) proteins within each bin corresponding to protein domains conserved between human and yeast. Dashed line indicates the boundary between proteins with or without protein domains present in both yeast and human proteomes. Inset (top right) shows density of interactions within two subspaces containing proteins with or without conserved domains, respectively.

C   Domain-specific interaction propensity measured by the degree ratio ($k_{inter-}/k_{intra-}$) of individual human or yeast proteins containing each protein domain. Proteins rank ordered by the ratio of their inter-species over intra-species degrees (left panel, magenta bars). Magenta lines (three right panels) indicate the rank of human or yeast proteins containing the domains (indicated on top) associated with significantly higher inter-species interaction propensities. Empirical *P*-values obtained by comparing to 10,000 randomized network controls for the three domains are: WD40, 0.02; zf-C3HC4, 0.04; Thioredoxin, 0.02.

first-of-its-kind verification of that concept by decoupling biophysical interaction properties from functional characteristics. Correlations of degree with pleiotropy for yeast proteins are greatly diminished within the inter-interactome (Fig 5C). This finding suggests that in spite of common network topological characteristics, coordination between the biophysical interactions and function is fundamentally altered in the inter-interactome.

**Correspondence between the inter-interactome and intra-species functional networks**

We next examined overlaps of the inter-interactome with functional features assigned to yeast genes. Yeast functional attributes and not human were selected because functional annotations and systematic functional genomic maps are more readily available for

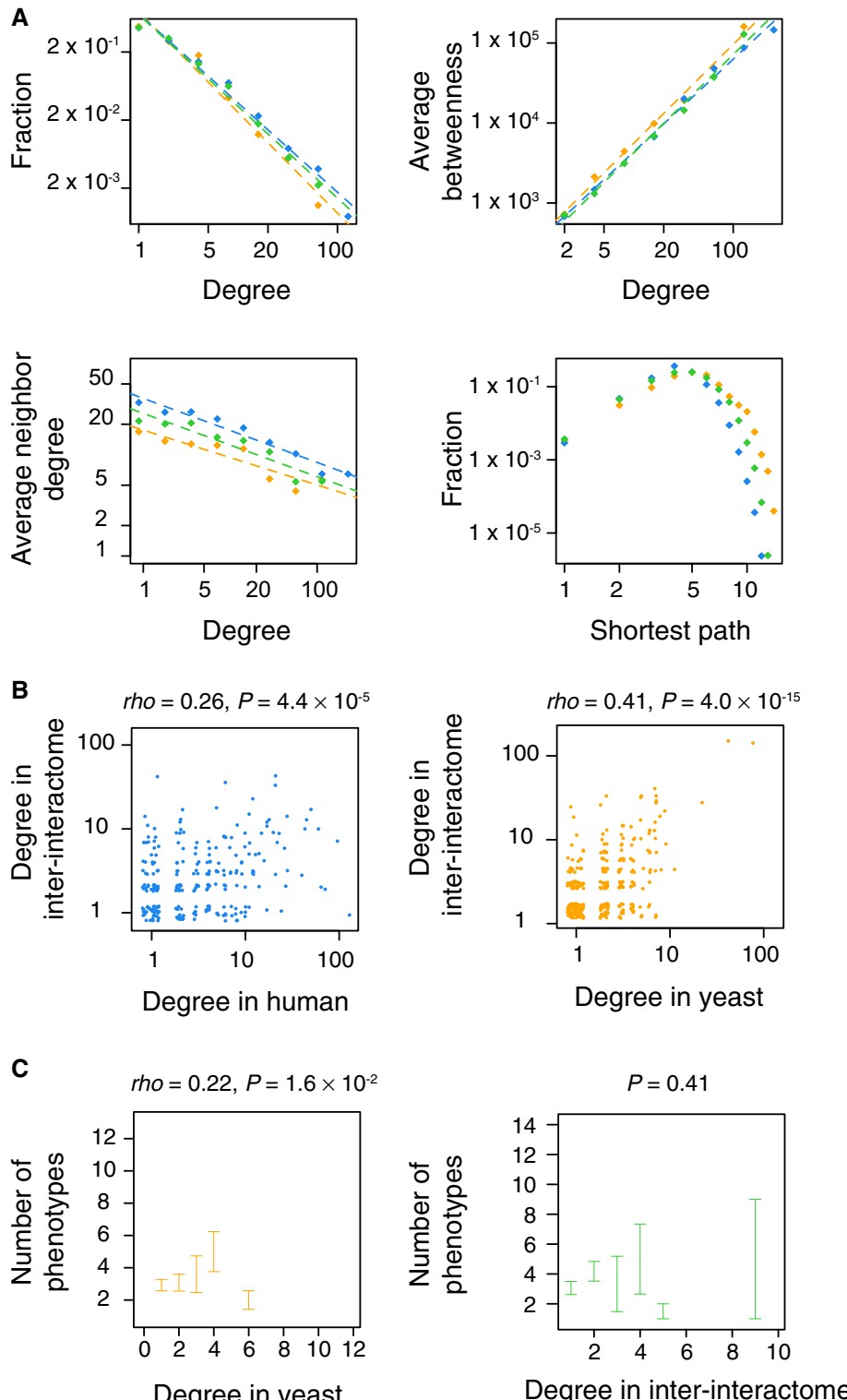

**Figure 5. Global network properties of the human–yeast inter-interactome.**

A Topological properties for the inter-interactome (YHII-1, green dots) compared to the human (HI-1, blue dots) and yeast (YI-1, orange dots) networks.

B Human (left) and yeast (right) proteins have correlated degrees in the inter-interactome (YHII-1) and human (HI-1) or yeast (YI-1) networks. Spearman's correlations (rho) are shown.

C Numbers of phenotypes associated with deletion of the encoding yeast gene at indicated degree in the yeast (YI-1, left) and inter-interactome (YHII-1, right) data sets. Orange (left) and green (right) dots are averages of black dots at each degree cutoff. Error bars indicate standard error of the proportion. *P*-values are for Pearson's correlation test.

yeast. To allow comparison of pairs of yeast proteins connected by a human protein in the inter-species network to pairs of yeast proteins in the intra-species network, we first determined levels of functional overlap for pairs of yeast proteins located two degrees of separation from each other in the yeast intra-species network. As expected, co-functionality levels were not as high as for directly interacting yeast proteins (Gunsalus *et al*, 2005). For all three co-functionality measures used, shared Gene Ontology (GO) (Ashburner *et al*, 2000), Pearson's correlation coefficient (PPC) measurements of synthetic lethality profile similarities (Costanzo *et al*, 2010), and co-expression similarities (Yu *et al*, 2008), the levels of overlap between biophysical interactions and co-functionality were significantly reduced in the inter-interactome network relative to the yeast intra-species network (Figs 6A and B, EV1 and EV2).

The overlap between inter-species biophysical and intra-species functional networks revealed significant "remnants of co-functionality" as compared to random expectation by all three functional indices examined (Figs 6A and B, EV1 and EV2). At various thresholds of GO term specificity, for example, inter-species biophysical interactions were up to 10-fold more likely to be co-functional than what would be expected by chance (Fig 6A). Such enrichment of GO annotations among inter-species interacting protein pairs remains significant upon removal of paralogs from the network (Fig EV1). Non-conserved human proteins appear to mediate functionally meaningful inter-species interactions. For example, neither MCMBP nor SMN2 have readily detectable homologs in yeast, but both proteins physically interact with pairs of yeast proteins that have closely related functions (Fig 6A).

These observations uncover remnants of co-functionality between yeast and human proteomes, despite substantially perturbed coordination between inter-species interactions and intra-species co-functionality.

### Inter-connected communities in the inter-interactome and the two parent networks

In biophysical interactome maps, functionally related proteins tend to form highly connected network "cliques" or "communities" (Barabási *et al*, 2011; Vidal *et al*, 2011). To investigate how remnants of co-functionality might be globally organized in the inter-interactome network, we used a link-clustering method (Ahn *et al*, 2010; *Arabidopsis* Interactome Mapping Consortium, 2011) to identify communities of densely clustered interactions with significant levels of GO enrichment (Fig 7A and Table EV6). The fraction of inter-species communities that are enriched for shared GO terms is significantly higher than that of randomized controls, similar to what is observed for the two parent intra-species networks (Fig 7B). Among the 392 inter-species interactions in GO-enriched communities found in the inter-interactome (Table EV6), 292 (~70%) involve either yeast or human non-conserved proteins. Hence, co-functionality seems common and percolates throughout the inter-species network, involving non-phylogenetically conserved proteins.

There are many communities containing non-conserved proteins in the inter-interactome. One example community contains Atg8 (Fig 7A), a yeast protein essential for autophagy. Autophagy is a conserved eukaryotic pathway for sequestering and transporting cytoplasmic and organellar proteins to the lysosome for degradation (Shpilka *et al*, 2011). Atg8 interacts with six human proteins, three of which (BNIP3, BNIP3L, MLX) share the functions in apoptosis and immune responses. Two of the six human interactors (BNIP3L, TBC1D5) are known to interact with the human homologs of Atg8 (Rual *et al*, 2005; Novak *et al*, 2010; Popovic *et al*, 2012; Rolland *et al*, 2014). Neither BNIP3 nor BNIP3L has a yeast homolog. This inter-species community suggests a route by which species-specific functions mediated by non-conserved proteins are coupled to highly conserved and ancient cellular machineries.

Since distinct GO-enriched communities may overlap and share multifunctional proteins (Ahn *et al*, 2010), we tested how intra-species parental and inter-species inter-interactome network communities might relate to each other by linking them through proteins that belong to at least two distinct, intra-yeast, intra-human, or inter-species communities. These linkages give rise to a significantly connected network of communities, relative to random controls (Fig 7C), suggesting that remnants of co-functionality and intra-species co-functionality are highly inter-related. Several inter-interactome and yeast YI-1 communities share proteins and are enriched for functions related to protein trafficking (Fig 7A), a cellular process highly conserved between yeast and human (Wickner & Schekman, 2005).

## Discussion

Inter-species protein–protein interactions have been mapped (Diss *et al*, 2013) to study pathogen–host interactions (Calderwood *et al*, 2007; Selbach *et al*, 2009; Mukhtar *et al*, 2011; Jager *et al*, 2012; Pichlmair *et al*, 2012; Rozenblatt-Rosen *et al*, 2012), to identify evolutionary modifications of protein binding specificity (Zarrinpar *et al*, 2003; Zamir *et al*, 2012; Das *et al*, 2013), and to characterize chimeric protein complexes in hybrid cells of closely related species (Leducq *et al*, 2012; Piatkowska *et al*, 2013). The human–yeast inter-interactome presented here is unique in that it is the first inter-species interactome mapped between two evolutionarily distant proteomes. The systematic mapping strategy, the coverage at proteome scale, the inclusion of proteins that do not have homologs in the opposing species, and the well-controlled experimental conditions matching the inter-interactome to two intra-species parent networks have allowed us to analyze the commonalities and differences between inter- and intra-species networks, gaining insights into the evolution of biological networks and ancestral gene function.

### Inter-interactome mapping as a new way to study network evolution

How biological systems evolve toward complexity and diversity is a central question (Carroll, 2001). Studies on the evolution of cellular networks have primarily focused on understanding how interactions vary over evolutionary time periods between phylogenetically conserved proteins (Sharan & Ideker, 2006). Critically unresolved are two fundamental issues: (i) how new interactions arise and (ii) how biophysical and functional interactome networks remain coordinated over evolutionary time (Fig 1).

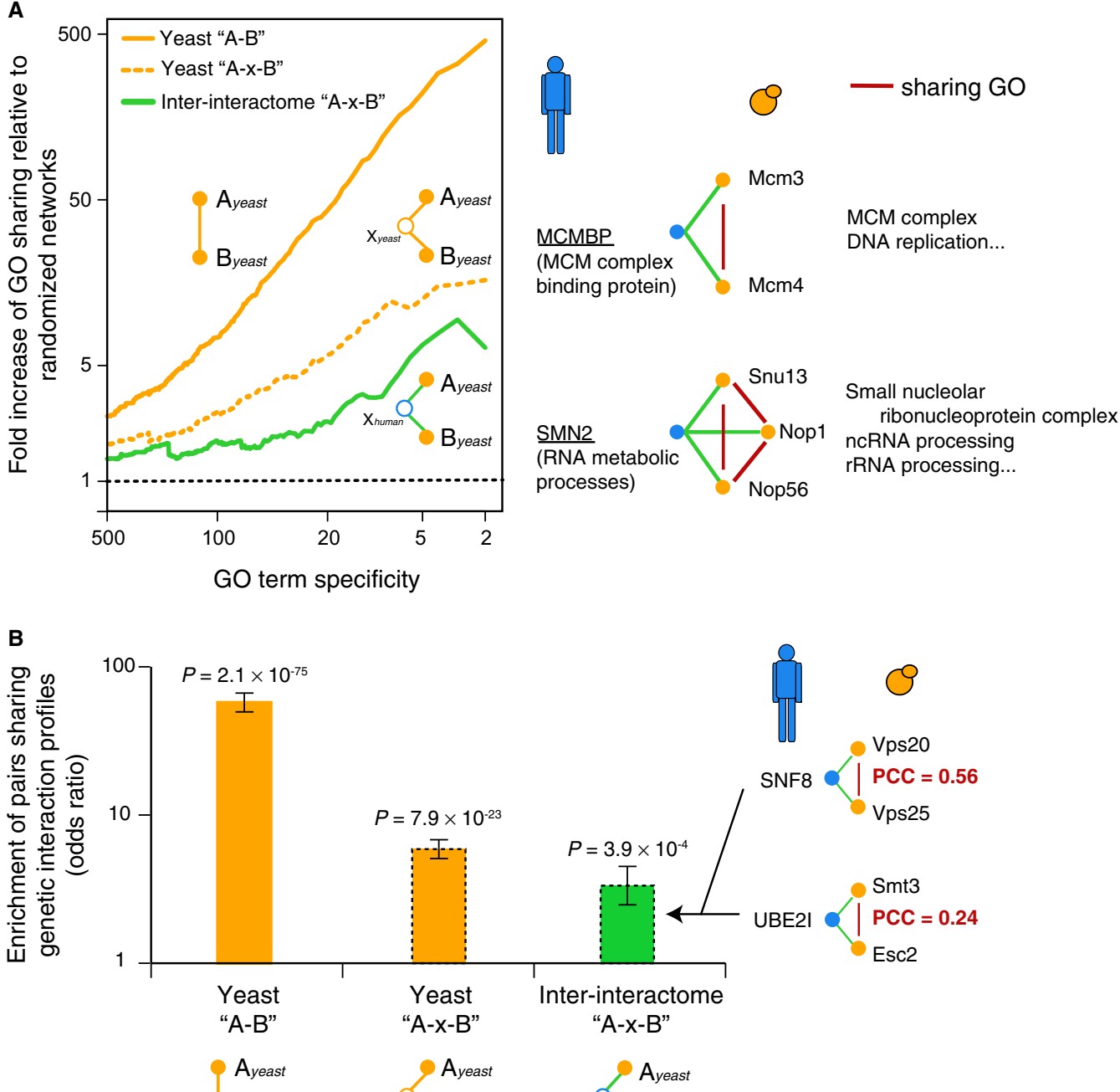

**Figure 6.  Remnants of co-functionality in the human–yeast inter-interactome.**

A  Fold increase of interacting yeast protein pairs sharing specific GO in the inter-interactome (YHII-1, green) and intra-species yeast (YI-1) networks relative to randomized network controls, at indicated cutoffs of GO specificity (left). Inter-species example pairs are shown with two non-conserved human proteins underlined.

B  Odds ratio enrichments of interacting yeast protein pairs with genetic interaction profile Pearson's correlation coefficients (PCC ≥ 0.2). Error bars indicate standard error. *P*-values of enrichment determined by Fisher's exact test. Inter-species example pairs are shown.

Our findings that human–yeast inter-species interactions uncoupled from direct selective pressure are as prevalent as intra-species interactions (Fig 4) support the existence of "pseudointeractions" (Venkatesan *et al*, 2009), *that is,* interactions between proteins with little or no functional relationships. Some non-functional interactions may be as robustly detectable as human–yeast inter-species interactions, while others may be relatively low-affinity interactions occurring at much higher frequency (Deeds *et al*, 2007; Zhang *et al*, 2008). Such functionally insignificant interactions (Levy *et al*, 2009; Venkatesan *et al*, 2009; Gray *et al*, 2010), if not

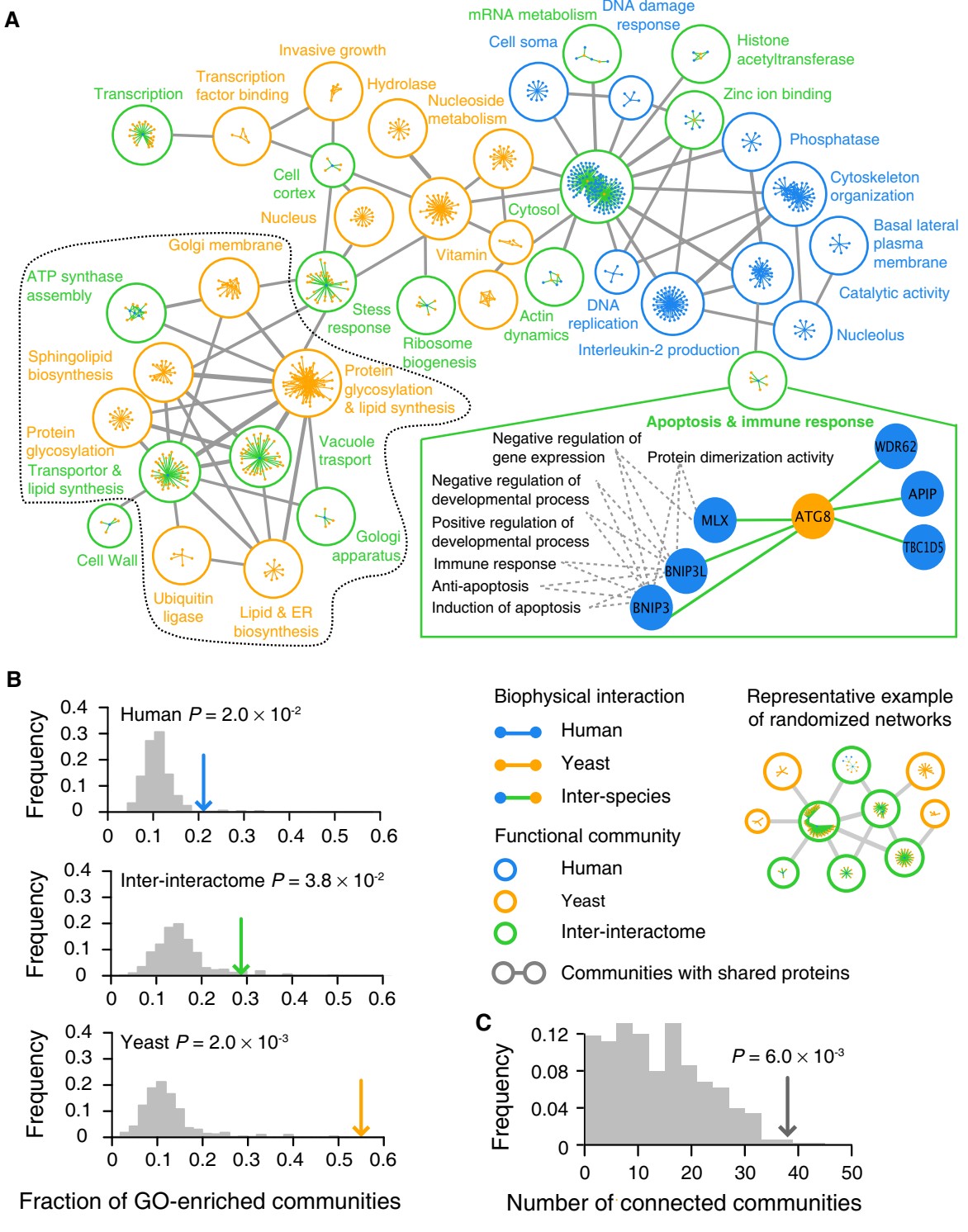

**Figure 7.  Inter-connected network communities.**

A   Function-enriched network communities in the inter-interactome (YHII-1) and intra-species human (HI-1) and yeast (YI-1) networks. Communities are connected if they share common protein nodes, with line thickness corresponding to the number of nodes shared between them. Unconnected communities are not shown. Each community is labeled with representative enriched function. A community containing Atg8 is enlarged showing several representative GO annotations enriched in this community. Communities within the dashed black curve (bottom left) are enriched for functions related to protein trafficking.

B   Fraction of GO-enriched communities (arrows) in the human (HI-1), inter-interactome (YHII-1), and yeast (YI-1) networks compared to distributions of randomized network controls (gray histogram). A representative example from randomized networks is shown. Empirical *P*-values = 0.002.

C   The total number of linked communities sharing common proteins from the human (HI-1), inter-interactome (YHII-1), and yeast (YI-1) networks (gray arrow) compared to distributions of randomized network controls (gray histogram). Empirical *P*-value = 0.002.

purged by purifying selection or lost due to genetic drift (Fernandez & Lynch, 2011), may loosen the correspondence between biophysical interaction and functional networks.

Several protein domains exhibit reduced interaction propensity within species relative to between species (Fig 4C). Combined with evidence that non-functional interactions may be more frequent in the inter-interactome than within species networks (Fig 6), this finding supports opposing evolutionary forces on biological networks. Natural selection acts not only to retain functional interactions, as is widely appreciated, but also to remove deleterious interactions (Zarrinpar *et al*, 2003), strengthening the correspondence between biophysical and functional networks.

The resemblance between the inter-interactome and the two intra-species parent networks (Figs 4 and 5) argues that, instead of being strictly adaptive, global topological features of biological networks might at least in part reflect intrinsic properties of proteins.

### Inter-interactome mapping as a new way to study gene function

Vastly different organisms can be related to one another genetically through the presence of conserved genes in their genomes, reflecting the common origin of living organisms and evolutionary constraints on gene function. Ancestral function in proteins is frequently inferred through cross-species comparisons. Cross-species genetic complementation experiments (Dolinski & Botstein, 2007; Kachroo *et al*, 2015) indicate conserved function between orthologs. The molecular basis of such complementation is often left uncharacterized. Comparative interactomic approaches reveal cross-species conserved molecular interactions, but cannot readily identify conserved binding specificity in proteins, as co-evolved interacting protein pairs may alter ancestral binding capabilities (Zamir *et al*, 2012).

Mapping inter-species interactions between distant proteomes has identified overlapping interactors between distantly related homologous protein pairs (Figs 2A and 3A, and Table EV3). Such inter-species molecular complementarities in proteins may correspond to evolutionarily conserved ancestral binding sites and underlie cross-species functional complementation. Proteins that interact with both human–yeast homologs do not necessarily themselves to be conserved between human and yeast (Fig 2A and Table EV3), suggesting that conserved proteins may mediate species-specific interactions. Non-conserved proteins form network communities enriched for conserved or species-specific functions (Figs 6 and 7, and Table EV6).

How could non-conserved proteins form interactions at ancestral binding sites? First, species-specific gene loss (Koonin, 2003) may lead to conserved proteins interacting with proteins maintained in one species but lost in the other (Kim *et al*, 2006). Second, some proteins, due to their structural modularity (Chothia *et al*, 2003), may undergo gross sequence variations yet still preserve certain ancestral structural features required for specific interactions. Third, conserved proteins may acquire new functions with species-specific new proteins through conserved binding sites (Zeke *et al*, 2015).

Exploitation of ancestral binding sites for new functions is consistent with gene co-option in evolution (True & Carroll, 2002), when natural selection finds new functions for existing genes. To account for how biological systems evolve toward complexity and diversity, various models have been put forward, which focus on changes in genes and gene products as well as their regulations (Carroll, 2008; De Robertis, 2008). The appearance of new interactions in proteins also involves modifications of pre-existing binding interfaces (Bridgham *et al*, 2006; Ernst *et al*, 2009), or formation of new interfaces (Fernandez & Lynch, 2011). Countering evolutionary changes, proteins may need to retain specific ancestral properties due to pre-existing functional constraints. Biological networks may evolve without gross changes in proteins, when ancestral binding sites acquire new interactions. New interactions at conserved binding sites link ancestral cellular machineries to new species-specific functional modules in complex biological systems. Just as sequence conservation indicates crucial functional constraints on genomes, inter-interactome mapping may reveal crucial functional constraints on biological networks.

## Materials and Methods

### Interactome mapping

Interactome mapping was carried out essentially as described (Dreze *et al*, 2010), but with four modifications to the pipeline. (i) A single yeast two-hybrid (Y2H) configuration was used; the configuration with yeast proteins expressed from the pDEST-DB vector and human proteins from the pDEST-AD vector. (ii) DB-X and AD-Y constructs were separately PCR amplified from lysates of the first-pass Y2H-positive colonies and then were PCR stitched together, and then stitched PCR products were pooled and sequenced by next-generation sequencing on a Roche 454 FLX platform (Yu *et al*, 2008). (iii) Reproducibility of the Y2H interaction phenotype was assessed by pairwise testing each first-pass pair four independent times, each time retrieving fresh clones from glycerol stocks, and retaining only those interactions that scored positive all four times. (iv) Each interacting pair was confirmed by Sanger end-sequencing of PCR amplified DB-Xs and AD-Ys from the final retained Y2H-positive clones. Detailed description of all experiments is provided in the Appendix Supplementary Methods.

### Validation of interacting pairs by LUMIER with BACON

We randomly selected 160 yeast and human protein pairs from the 1,583 verified pairs (the inter-interactome YHII-1 test set). We also selected at random 200 pairs from the Y2H search space to constitute a random reference set (RRS) of negative controls. Each ORF selected was available as single-colony isolated, full-length sequence-verified Entry clones (Yang *et al*, 2011). For each pair, an interaction between the gene products was tested with a modified LUMIER assay (LUMIER with BACON) (Taipale *et al*, 2012). We also tested by LUMIER a subset of the previously compiled human protein–protein interaction reference sets (Venkatesan *et al*, 2009) for which we have available single-colony isolated, full-length sequence-verified Entry clones (Yang *et al*, 2011).

LUMIER with BACON assay was carried out as before (Taipale *et al*, 2012). All ORFs are cloned by Gateway recombinational cloning into vectors based on the pcDNA3.1 plasmid, which carries N-terminal 3×FLAG and V5 tags or the N-terminal *Renilla* luciferase

tag. Transfection to 293T cells is carried out in 96-well plate format. Two days after transfection, cells are washed in 1 × PBS and lysed in HENG buffer (50 mM HEPES-KOH pH 7.9, 150 mM NaCl, 2 mM EDTA, 5% glycerol, 0.5% Triton X-100 supplemented with protease, and phosphatase inhibitors). An aliquot of the cell lysate is transferred to 96- or 384-well plates (Greiner). Luminescence in each well is measured on an Envision plate reader (Perkin-Elmer) using a Gaussia FLEX luciferase kit (New England Biolabs). This measurement gives luminescence in the total cell lysate ($L_{Total}$). The rest of the lysate is transferred to 96- or 384-well plates coated with anti-FLAG M2 antibody (Sigma-Aldrich). Plates are incubated at 4°C for 3 h, after which plates are washed with HENG buffer on an automated plate washer (Biotek). Luminescence in each well is measured using a Gaussia FLEX luciferase kit. This measurement gives luminescence in the immunoprecipitated fraction ($L_{Pull-down}$). After measurement of luminescence, HRP-conjugated anti-FLAG antibody in ELISA buffer (1 × PBS, 2% goat serum, 5% Tween-20) is added to the wells. One hour later, plates are washed in 1 × PBS/0.05% Tween-20 on a plate washer as before. ELISA signal is detected with TMB substrate (Thermo Pierce Scientific), and the absorbance (optical density) is read at 450 nm. Each plate contained eight wells with a twofold dilution series of 3×FLAG-tagged *Gaussia princeps* luciferase. These control wells provide normalization and set a standard for ELISA signal. We calculated an immunoprecipitation percentage (IP%) for each tested well (IP% = $L_{Total}/L_{Pull-down}$). Non-specific binding of *Renilla*-tagged protein to the well or the binding of *Renilla* tag to the Flag-tagged protein causes background luminescence that needs to be subtracted. To that end, we repeat the interaction assay with two controls, cells transfected with only *Renilla* luciferase tagged protein (control R) or co-transfected with FLAG-tagged protein with *Renilla* luciferase (Control F). We consider pairs that meet the following criteria:

(1)    $IP_i\% > 0$
(2)    $IP_{i\_control\_R}\% > 0$
(3)    $IP_{i\_control\_F}\% > 0$
(4)    ELISA signal is above background level ($\geq 0.07$)

Interaction scoring is based on the difference between IP% in the testing wells containing both the bait and prey protein versus the maximum IP% from the two corresponding controls, Control R and Control F.

$$Score = IP_i\%/Max\left[IP_{i\_control\_R}\%, IP_{i\_control\_F}\%\right]$$

Each pair X-Y (*i.e.,* proteins X and Y) was tested in two LUMIER configurations: with X-FLAG tag and Y-*Renilla* luciferase fusions, as well as Y-FLAG tag and X-*Renilla* luciferase fusions. Each configuration was tested three times with independent transfections each time. Only those pairs where three valid LUMIER readings appeared for each configuration were used for interactome quality assessment. The average score of the three repeats is calculated. A combined LUMIER score for a given pair X-Y is the higher of the average score of the two configurations (Table EV7). In the bar plot (Fig 2C), error bars represent the 95% confidence intervals and were computed by modeling the recall at any point as a Bernoulli process with standard error of the mean (SQRT[$p \times (1-p)/n$]), where "$p$" is the recall at that threshold, and "$n$" is the number of pairs in the

subset being examined). *P*-values were determined by chi-square test with Yates correction. All analyses were carried out in the R computing environment (R Development Core Team, 2009).

## Assembly of the systematic inter- and intra-species interactome networks

The yeast–human inter-interactome YHII-1, containing 1,583 pairs of interacting yeast–human proteins is bipartite, with yeast proteins tested only in the DB configuration and human proteins tested only in the AD configuration (Table EV2). To match the DB-X ORF search spaces with the inter-interactome, we removed interactions from yeast (YI-1) (Yu *et al*, 2008) that were tested with 5 mM 3-AT. This yields a YI-1 network containing 1,690 interactions. The human (HI-1) network (Rual *et al*, 2005) was updated with Entrez Gene ID downloaded on August 2, 2010. ORFs that no longer mapped to the updated human gene models were discarded from the analyses. The updated HI-1 network contains 2,750 interactions. Homodimers were excluded for most analyses. There are a total of 1,641 and 2,611 heterodimeric interactions in YI-1 and HI-1, respectively.

## Homology assignments

We combined human–yeast homology mapping downloaded from SGD (Cherry *et al*, 2012), InParanoid 7 (Ostlund *et al*, 2010) and InParanoid 8 (Sonnhammer & Ostlund, 2015) to determine homology relationships between yeast and human proteins. To identify lineage-specific proteins among human and yeast non-homologs, we used assignments from the HomoloGene database (Wheeler *et al*, 2007), and identified human and yeast proteins with only metazoan or fungal orthologs, respectively.

## Overlap of inter-species and intra-species interactors

Common interactors shared by human–yeast homologs were identified by combining the inter-interactome YHII-1 and known intra-species interactions obtained from either of the parent systematic intra-species maps (Rual *et al*, 2005; Yu *et al*, 2008) (Fig 3A) or interactome maps downloaded from the Mentha database (Calderone *et al*, 2013) (Table EV3). In the calculation of the fraction of inter-species interactions in YHII-1 that revealed common interactors between yeast–human homologs, the denominators were the numbers of inter-species interactions involving proteins with homologs in yeast or human that were also in YI-1 or HI-1 search spaces, respectively. The observed overlaps between inter- and intra-species networks were compared to randomized network controls (Fig 3A).

## Network randomization schemes

For empirical statistical testing, we created randomized networks controlling for specific hypotheses. We employed a network randomization strategy in which the total number of interactions in the real network, the collection of possible proteins found for AD or DB configuration, and the number of opposing configuration interactors found for each protein when tested in either configuration were all preserved. Because the same edge might appear twice

in a given randomization, the algorithm was designed to re-scramble duplicated edges with a subset of the unduplicated edges. This randomization scheme was used for all analyses.

### Protein structural complexes

To analyze the structural basis of inter-species interactions, we considered yeast and human proteins that have three-dimensional structures in PDB (PDB release as of January 26, 2013). Mapping of yeast ORF IDs and human gene Entrez (Maglott *et al*, 2011) Gene IDs to Uniprot Accession Codes (ACs) (Uniprot Consortium, 2012) was achieved by considering SwissProt entries with taxonomy IDs 559292 (*S. cerevisiae*) and 9606 (*H. sapiens*) using the Uniprot mapping service (http://www.uniprot.org/?tab=mapping). Entries that were mapped to multiple Uniprot ACs were solved manually (Seal *et al*, 2011), whenever possible. The remaining multiple mappings were eliminated. Orthology mapping between yeast and human proteins used Inparanoid (standalone version 4.1) (Ostlund *et al*, 2010) with default parameters. For in-paralog clusters, we used the matrix expanded set of pairs to maximize coverage.

We took all available structures of protein complexes in PDB and used the mapping provided by SIFTS (Velankar *et al*, 2013) (http://www.ebi.ac.uk/pdbe/docs/sifts/). We eliminated from the data sets biological units containing non-protein chains, chains that could not be mapped to a Uniprot AC, and chains of less than 30 residues. For each PDB complex, we identified internal direct interactions between the components. We considered every chain as a single component, and filtered out PDB structures where one chain could be mapped to different proteins—that is, chimeric constructs.

We considered two components (chains) to be directly interacting if they share any residue–residue contacts among: (i) covalent interactions (disulfide bridges), defined as two sulfur atoms of a pair of cysteines at a distance $\leq 3.0$ Å; (ii) hydrogen bonds, defined as all atom pairs N–O and O–N at a distance $\leq 3.4$ Å; (iii) non-bonded interactions defined as all atom pairs N–O and O–N at a distance $\leq 4.0$ Å or all pairs of carbon atoms at a distance $\leq 4.5$ Å.

The yeast and human proteome sets are composed of 6,621 and 20,226 proteins, respectively. The orthology mapping calculated with standalone Inparanoid 4.1 is composed of 4,566 pairs between 3,701 human proteins and 2,425 yeast proteins. Of the total of 7,241 tested human genes, 7,115 (98.3%) could be univocally mapped to 7,106 Uniprot ACs in the human proteome. Of the total of 3,778 tested yeast ORFs, 3,698 (97.9%) could be univocally mapped to 3,696 Uniprot ACs in the yeast proteome. In the inter-interactome, 1,581 (99.9%) interactions could be mapped to 1,580 unique pairs of yeast and human proteins with Uniprot ACs.

We found a total of 17 of inter-species interactions for which the yeast or human orthologs can be mapped in the same structural protein complexes (Table EV4) nine in yeast complexes and 12 in human complexes, with an overlap of four found in interologous protein complexes in both organisms. Of each of the 17 inter-species interacting pairs, the yeast or human ortholog and the inter-species interactor were found to be in direct contact in the structural model. Enrichment was calculated by Fisher's exact test. Details are provided in the Appendix Supplementary Methods.

### Protein domain assignments and inference of domain–domain interactions

Protein sequences, translated from DNA sequences of individual ORF clones using Bioconductor, were processed by a local installation of InterProScan (Zdobnov & Apweiler, 2001) to assign Pfam domains for the ORFs in our collection. We downloaded domain–domain interactions annotated in the DOMINE database (Yellaboina *et al*, 2011) at (low, medium, high) levels of confidence. We identified interacting protein pairs that had matched interacting domain pairs as annotated in DOMINE at the relevant level of confidence (Table EV5). Such interacting protein pairs might be explained by domain–domain interactions. We plotted the fractions of interactions that could be explained by domain–domain interactions at high confidence in each of the three networks (YI-1, YHII-1 and HI-1) together with the distribution of such fractions in their corresponding randomized networks (Fig 3B). The denominators were interacting protein pairs in the subsets of the three networks for which both proteins contained at least one domain implicated in domain–domain interactions. Homodimers in the YI-1 and HI-1 networks were removed from this analysis for fair comparison with YHII-1.

### Measurements of shared interaction profile similarity

To measure the fraction of partners shared by two proteins, we used the Jaccard index or Jaccard similarity coefficient, defined as the size of the intersection divided by the size of the union of the sample sets. Here, the Jaccard index corresponds to the number of shared interactors divided by the total number of interactors. Self-interactions, or homodimers, were not included in these counts. We compared the Jaccard interaction similarities of yeast–human homologous pairs, non-homologous yeast–human protein pairs sharing a common protein domain, and other pairs of yeast–human protein pairs sharing at least one interacting partner. The Mann–Whitney *U*-test was used to make pairwise comparisons between distributions of interaction profile similarities (Fig 3C).

### Proteome-wide distribution and density of inter-species interactions

The YHII-1 inter-interactome was mapped testing yeast proteins in the Y2H DB configuration and human proteins in the AD configuration. For fair comparison with the intra-species networks, in which most proteins were tested in both the DB or AD configurations, the interaction density was defined as the average number of unique DB-X and AD-Y interacting pairs detected per screen divided by the total number of unique DB-X and AD-Y pairs in a given search space (Fig 4A and B). The number of unique DB-X and AD-Y interacting pairs per screen is the same as the total number of detected edges in the inter-interactome, as inter-interactome pairs were screened in only one configuration. But this is not true for the intra-species networks, in which some interactions were found in both Y2H configurations or in multiple screens, yielding a network with higher saturation. This was controlled for in the human (HI-1) network (Rual *et al*, 2005), which was screened once in both configurations. For the yeast (YI-1) network (Yu *et al*, 2008), which was screened three independent times in both configurations, interaction densities

were calculated by averaging the number of unique DB-X and AD-Y interacting pairs from each independent screen.

For matrix representation of the inter-interactome (Fig 4B), green dots represent interactions between yeast and human proteins. Proteins were binned and arranged according to the percentage of their sequence within domains found in both species (human, 50 bins, left; yeast, 25 bins, top). Thresholds are described along the axis, and the dashed black line delineates the threshold at which proteins have no domains found in both yeast and human proteins.

### Domain-specific interaction propensity in intra- and inter-species networks

For each yeast or human protein that had their respective DB or AD configuration-specific degree above zero in both inter- and intra-species networks, we calculated the ratio between that protein's configuration-specific degree in the inter-interactome and in the intra-species network ($k_{inter}/k_{intra}$). To control for different interaction coverage of the inter- and intra-species networks, we normalized the ratio ($k_{inter}/k_{intra}$) by the average degrees of the inter-interactome and the intra-species network (($k_{inter}/k_{intra}$)/(Average $k_{inter}$/Average $k_{intra}$)). We ordered the yeast DB-X baits and human AD-Y preys based on their normalized Inter-/Intra-degree ratios. The specific distributions of the normalized Inter-/Intra-degree ratios for sets of proteins with and without particular domains were compared using a Mann–Whitney *U*-test. For empirical statistical comparisons, we created 10,000 randomizations using the following strategy. We preserved the network structures of the inter-interactome and the intra-species yeast and human networks. Protein nodes were binned by the sum of their normalized degree in the inter-interactome and intra-species networks ($k_{total} = (k_{inter}$/Average $k_{inter}$) + ($k_{intra}$/Average $k_{intra}$)). Bins included proteins with $k_{total}$ of 1, 2, 3–4, 5–8, 8–16, 17–32, 33–64, 65–128, and 129–256. We permuted the domain annotation vectors among proteins in the same bin. This strategy was designed to preserve the general domain annotations of the proteins in the three networks, the correlation between $k_{inter}$ and $k_{intra}$ (Fig 4C), the potential correlation between specific domains and $k_{total}$, and eliminated the excess $k_{inter}/k_{intra}$ that some domains may have. Empirical *P*-values were obtained by comparing the *P*-values by the Mann–Whitney *U*-test for each domain in the observed case and to that of the 10,000 randomized situations. Homodimers in the YI-1 and HI-1 network were removed from this analysis for fair comparison with the inter-interactome.

### Prediction of disordered residues

Disordered residues for each ORF were predicted using a local installation of the Disopred2 program (Ward *et al*, 2004). Disordered fractions of residues were collapsed to the gene level by calculating the average at the ORF level.

### Network topological measures

Degree distributions, betweenness as a function of degree, disassortativity (average degree of neighbors as a function of degree) and shortest path distribution plots (Fig 5A) were calculated within the main component of the three networks (YI-1, YHII-1 and HI-1) after

removal of homodimers. The "igraph" package for R (R Development Core Team, 2009) was used to perform these calculations for nodes or sets of nodes within each of the networks. Although the clustering coefficient was not compared, because the bipartite nature of the inter-interactome allows for no closed triangles, the small-world property of networks is often considered to be characterized by high clustering coefficients as well as short path lengths (Yu *et al*, 2008).

### Correlation between inter- and intra-species network degrees

To see whether the same proteins engage in similar numbers of interactions within or between species, we tested whether matched configuration-specific degree counts in the intra-species network correlated with those in the inter-interactome. Configuration-specific degree counts are the counts of unique interactors detected in a particular configuration in Y2H. Configuration-specific degree counts differ from regular protein network degrees in that they reflect the propensity of a protein to form interactions in a particular Y2H configuration rather than in the combination of both. The inter-interactome was mapped in a single Y2H configuration, with yeast proteins tested in the DB configuration and human proteins tested in the AD configuration. Degree and configuration-specific degree of proteins in the inter-interactome are identical. But this is not true for the intra-species yeast (YI-1) (Yu *et al*, 2008) and human (HI-1) (Rual *et al*, 2005) networks, in which most proteins were tested in both configurations. The majority of interactions were detected in one but not the other Y2H configuration. We plotted and tested for the presence of a significant Spearman's correlation between counts of AD interactors found for the same yeast protein tested in DB configuration in YI-1 and the inter-interactome, as well as the correlation between counts of DB interactors found for the same human protein tested in AD configuration in HI-1 and the inter-interactome (Fig 5B). Spearman's correlation significance was determined using R (R Development Core Team, 2009). Homodimers were removed from this analysis because such interactions cannot exist in an inter-species bipartite network.

### Correlation between network degree and pleiotropy

The relation between protein connectivity and pleiotropy was tested as previously described (Yu *et al*, 2008). Phenotypic profiling data sets were downloaded (Dudley *et al*, 2005). We removed the most highly correlated data sets among the 22 conditions of phenotypic profiling data and used the remaining 19 independent conditions in our analysis. We compared the matched configuration-specific degree counts for yeast proteins tested in the DB configuration in the YI-1 network to those of the inter-interactome (Fig 5C), removing homodimers from YI-1 for fair comparison. Correlation between degree and pleiotropy was detected when considering yeast protein configuration-specific degree for YI-1 DB configuration data. No correlation between degree and pleiotropy was observed in the inter-interactome.

### GO enrichment analysis

The profile of GO annotation (Ashburner *et al*, 2000) assignments for each gene was extracted from the GO.db BioConductor R package

(R Development Core Team, 2009). Only GO annotations associated with "EXP", "IDA", "IMP", "IGI", "IEP", "ISS", "ISA", "ISM", or "ISO" evidence codes were used. To prevent circularity due to the inclusion of the previously published yeast (YI-1) (Yu *et al*, 2008) and human (HI-1) (Rual *et al*, 2005) networks, the "IPI" evidence code (Inferred from Physical Interaction) was excluded. GO annotation for each protein was propagated upward so that proteins were annotated with parent terms as well. For the sets of genes within each search space, GO annotation specificity for each GO annotation was defined as the number of genes in the search space with that GO annotation. Biological process, molecular function, and cellular component branch root terms ("GO:0008150", "GO:0003674", "GO:0005575", and "all") were not included.

Homodimers in the yeast YI-1 network were removed to derive the yeast YI-1 direct binary interaction "A-B" network. Unique protein pairs indirectly linked in the yeast YI-1 network through another yeast protein were compiled to derive the yeast (YI-1 "A-x-B") network. Unique protein pairs indirectly linked in the inter-interactome YHII-1 through a human protein were compiled to derive the inter-interactome (YHII-1 "A-x-B") network.

For each protein pair, the GO annotation profiles of the protein-coding genes were compared, and the level of the most specific shared GO annotation was assigned as the GO specificity for the protein pair. The smaller the GO specificity number, the more specific the shared GO annotation is. Cumulative fractions of the protein pairs with assigned GO specificity not greater than a range of GO specificities were calculated for all, and in 1,000 randomized networks. The fraction of protein pairs in the three networks having each possible level of specificity was divided by the fraction of pairs in 1,000 randomized networks having the same level of specificity (Fig 6A). We considered proteins that had at least one GO annotation with specificity less than 500. DB-X to AD-Y configuration-preserving networks were used, and homodimers in the YI-1 and HI-1 network were removed for fair comparison with the inter-interactome. Network randomization was carried out as described above.

To remove any confounding effects from gene duplication in yeast, we measured enrichment of GO annotations after removing paralogs from the yeast YI-1 and inter-interactome YHII-1 datasets. We identified yeast paralogs using the InParanoid (Ostlund *et al*, 2010; Sonnhammer & Ostlund, 2015) and HomoloGene databases (Wheeler *et al*, 2007). If two yeast genes are in the same group or cluster in any of these databases, they are considered to be a paralog pair. We measured GO enrichments at four specificity cutoffs (5, 20, 100, and 500) (Fig EV1).

### Genetic interaction correlation

Correlation of yeast genetic interactions (Costanzo *et al*, 2010) was downloaded from http://www.utoronto.ca/boonelab/data/public/sgadata_costanzo2010_correlatios.txt.gz. Proteins that were not found in the corresponding yeast (YI-1) or inter-interactome were removed from the analysis. The enrichment of interacting pairs with genetic interaction Pearson's correlation coefficients (PCC ≥ 0.2) is expressed as an odds ratio relative to non-interacting pairs (Fig 6B), in the three networks (YI-1 "A-B"), (YI-1 "A-x-B"), and (YHII-1 "A-x-B") described above,

### Expression correlation

We compiled yeast expression data from the Stanford Microarray Database (SMD) (Marinelli *et al*, 2008; Yu *et al*, 2008). We performed Pearson's correlation tests on the three networks (YI-1 "A-B"), (YI-1 "A-x-B") and (YHII-1 "A-x-B") described above, noting pairs that had significant ($P < 0.05$) correlation or anti-correlation. Pairs that did not have significant correlation were omitted from the density plot and Mann–Whitney *U*-test comparisons of the distributions (Fig EV2).

### Identification of functionally enriched communities

The link-based clustering algorithm to identify network communities (Ahn *et al*, 2010) was applied to the three networks (YI-1, YHII-1, and HI-1) as well as three sets of 500 randomized networks corresponding to each of the three original networks. To evaluate whether these communities appeared to be functionally coherent, we tested for GO annotation enrichments in communities containing five or more proteins using the FuncAssociate JSON interface in Python (Berriz *et al*, 2009). We used a custom GO annotation dictionary matched to the Bioconductor GO annotations and limited to the evidence codes used for the GO specificity analysis. We used proteins in each of the three network spaces as the relevant reference sets relative to which enrichments were determined. The adjusted false-discovery rate cutoff was 10%. We used Cytoscape (Shannon *et al*, 2003) to graph the functionally enriched communities detected and removed interactions not belonging to any enriched communities (Fig 7A).

Fractions of communities enriched in GO annotations in each of the three networks were compared to their corresponding randomized networks, yielding empirical *P*-values (Fig 7B). A representative combination of the three randomized networks, each with fractions of enriched communities closest to the median, was chosen for comparison (Fig 7B). The GO-enriched communities in all three networks (YI-1, HI-1, YHII-1) were further analyzed at the level of the shared protein nodes between inter- and intra-species communities, which is significantly larger than that found in 500 randomized controls, yielding an empirical *P*-value (Fig 7C).

### Data availability

The protein interactions from this publication have been submitted to the IMEx (http://www.imexconsortium.org) consortium through IntAct (Orchard *et al*, 2014) and assigned the identifier IM-24995.

Expanded View for this article is available online.

### Acknowledgements

The authors would like to thank Y. Jacob, J. De Las Rivas, A.J.M. Walhout, and members of the Center for Cancer Systems Biology (CCSB) at the Dana-Farber Cancer Institute for critical reading of the manuscript. This research was supported by National Human Genome Research Institute (NHGRI) grants R01-HG006061 awarded to M.V., D.E.H., and M.E.C. and R01-HG001715 awarded to M.V., D.E.H., and F.P.R. Y.X. receives support from National Science Foundation (NSF) grant CCF-1219007, Natural Sciences and Engineering Research Council of Canada (NSERC) grant RGPIN-2014-03892, Canada Foundation for Innovation (CFI) grant JELF-33732, and Canada Research Chairs

　　　　　　　　　　　　　　　　　　　　　

Program. F.P.R. received support from the Canada Excellence Research Chairs Program, the Krembil Foundation, and the Ontario Research Fund. M.V. is a Chercheur Qualifié Honoraire from the Fonds de la Recherche Scientifique (FRS-FNRS, Wallonia-Brussels Federation).

## Author contributions

MV and QZ conceived the project. QZ performed most of the experiments and SJP carried out most of the computational analyses. MiT, XY, FM, PH, RRM, FG, ST, AM, AD, PR, and LG contributed reagents or provided laboratory assistance. QZ, TH, YW, RM, JM, MuT, CF, BS, and MEC contributed to or assisted computational analyses. SL, A-LB, DEH, PA, YX, FPR, and MV provided support. QZ and SJP interpreted results and wrote the manuscript with MEC, YX, FPR, and MV with input from MAC, TR, BC, SL, A-LB, DEH, and PA.

## Conflict of interest

The authors declare that they have no conflict of interest.

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
