## [Review Process File · Molecular Systems Biology]

An inter-species protein-protein interaction network across vast evolutionary distance

Quan Zhong, Samuel J. Pevzner, Tong Hao, Yang Wang, Roberto Mosca, Jörg Menche, Mikko Taipale, Murat Taşan, Changyu Fan, Xiping Yang, Patrick Haley, Ryan R. Murray, Flora Mer, Fana Gebreab, Stanley Tam, Andrew MacWilliams, Amélie Dricot, Patrick Reichert, Balaji Santhanam, Lila Ghamsari, Michael A. Calderwood, Thomas Rolland, Benoit Charlotiaux, Susan Lindquist, Albert-László Barabási, David E. Hill, Patrick Aloy, Michael E. Cusick, Yu Xia, Frederick P. Roth and Marc Vidal

Corresponding authors: Marc Vidal, Harvard Medical School; Frederick P. Roth, University of Toronto; Yu Xia, McGill University; Quan Zhong, Wright State University

Review timeline:

Submission date:	06 August 2015
Editorial Decision:	30 August 2015
Revision received:	28 December 2015
Editorial Decision:	25 January 2016
Revision received:	22 February 2016
Accepted:	04 March 2016

Editor: Thomas Lemberger

Transaction Report:

1st Editorial Decision

30 August 2015

Thank you again for submitting your work to Molecular Systems Biology. We have now heard back from two out of the three referees who agreed to evaluate your manuscript. Given the recommendations provided by these 2 reviewers, I prefer to make a decision now rather than delaying further the process. As you will see from the reports below, the referees find the topic of your study of potential interest. They raise, however, a series of concerns on your work, which should be convincingly addressed in a revision of the present work. The recommendations provided by the reviewers are very clear in this regard and refer to the need of a much deeper and more extensive analysis of the results and for clarifications in several aspects of the study.

We would also kindly ask you to include in the submission the full inter-interactome data as Dataset file and strongly encourage you to deposit the validated interactions to an appropriate public IMEx database. Please include a "Data availability" section at the end of Materials & Methods to specify where the data are available (including the Dataset files included in the submission).

REFEREE COMMENTS

Reviewer #1:

Summary:

This paper presents a new PPI dataset of ~7,000 human proteins screened for physical interactions with ~4,000 yeast proteins. The analysis characterizes the functional and network properties of the resulting bipartite network. The authors make several conclusions based on this study: they conclude that inter-species network properties are similar to those of intra-species networks in terms of density and degree distribution; orthologs tend to interact with their partner's interactors more than expected by chance; there is limited functional information in the inter-species network, though it is better than random; intra-species degree doesn't correlate to gene properties (e.g. essentiality and pleiotropy) as strongly as intra-species degree does. In general, this is a unique dataset, which will be a valuable resource for several follow-up studies. However, the manuscript could be significantly strengthened with refinements to specific analyses (see details below). Also, more in-depth discussion of the motivation for collecting and potential applications of these data would be helpful.

Major comments:

(1) Regarding the human-yeast rescue analysis (Fig. 1c): is the fraction of PPI conserved for human orthologs predictive of the ability to rescue? I'm guessing there are several instances of human proteins that didn't rescue their essential yeast orthologs for which the authors also have PPI screen data-- this could be used to contrast the 25% PPI conservation among the "rescuers". A related, but more general question: is the fraction of conserved interactions different for essential vs. non-essential orthologs?

(2) Regarding the results on interaction density: it's surprising (to me, at least) that the density was very similar in both the intra- and inter-species networks. I would have expected either selection for or against interactions in real networks, at least for some proteins. Along these lines, can the authors examine whether there are specific domains or domain-domain pairs that are enriched or depleted in the inter-species network relative to the intra-species network? If so, this could reflect selection for/against interactions with certain domains. This sort of result would deepen this section of the paper. A related question: I may have missed it, but was there a difference in interaction density between conserved-conserved, zerolog-conserved, or zerolog-zerolog pairs? (I guess this analysis is part of Fig. 3A, but the statistics don't seem to be summarized anywhere--see more specific comments on 3A below)

(3) The lack of correlation between inter-species degree and essentiality or pleiotropy is one of the more interesting results in the paper. However, I'd suggest laying out the statistics supporting this finding more clearly. First, why is the range of degree (x-axis) in the left panel of Fig. 3C twice as big as the right panel? Does this have to do with how many data are available in each bin? The caption doesn't have any info that might explain this. Also, I find correlations computed on binned, cumulative data misleading. For example, for the essentiality result, I'd suggest just doing a test on the degree distributions of the essential vs. non-essential genes in the two networks (e.g. a Wilcoxon rank-sum test would be appropriate for this). For the "number of phenotypes" result, I would just suggest computing a correlation (either Pearson or Spearman) on the actual degree and the number of phenotypes for each gene (not the cumulative data that's plotted). If correlations computed using this more standard approach are consistent with the current findings, this would strengthen my confidence in this result.

(4) Related to the results presented in Fig. 2C: the homology-dependent overlap is a nice result. A related suggestion: can the authors check whether the overall conservation rate of a given protein's interaction neighbors is a predictive feature of the inter-species interaction frequency? For example, for orthologs gH and gY (for human and yeast), if 80% of gY's intra-species interactions are conserved in the human proteome, are there more likely inter-species interactions than if only 20% of gY's interactors are conserved in human? The answer to this would be a useful way of addressing the authors' query about the "extent to which inter-species interactions arise from evolutionarily conserved protein-binding mechanisms" since it is the intra-species interactions that would cause the conservation pressure.

(5) Comment on Figure 3A: It is somewhat difficult to understand the setup of this figure. I understand that the blue and yellow bar charts are functioning as x-axis labels for the adjacent histograms, but this could be labeled or stated in the legend. However, it isn't clear what is being measured along this x-axis. The main text ("the frequency of inter-species interactions between human and yeast proteins is independent of their levels of sequence conservation") is vague but seems to refer to conservation between sequences of orthologous pairs. However, the figure caption ("proteins are arranged and binned according to the fraction of their sequences found within protein domains present in both yeast and human proteomes") seems to say that a species-level evolutionarily conserved domain, not necessarily any ortholog, is enough to constitute conservation. Which is correct? And could the authors clarify in both locations? What do the dashed lines represent? Also, I'm not sure what's actually being plotted in the two plots referred to as histograms- I would expect something to sum to one here (either in each bin or across all bins, but that doesn't seem to be true). I think this figure could be simplified and all axes/legend labeled more clearly. It would be interesting to know if there is a difference between degrees of zerologs and orthologs, and it's hard to tell if the authors address this.

(6) Regarding the findings in this section: "Correspondence between the inter-interactome and intra-species functional networks", can the authors explain in more detail how they calculated the y-axis on all of these plots in Methods/figure legends (Fig. 4A, EV5). The fold enrichment they are plotting is quite sensitive to the background gene set, which I'm guessing varies between their inter-species and intra-species networks. Also, I noticed that two examples contain yeast duplicate genes (MCM3/MCM4 and SAM1/SAM2). In how many cases does "co-functionality" co-occur with sequence/structural similarity of the proteins? Could intra-species duplication be a confounding factor in the "remnants of co-functionality" result? (e.g. a human protein interacts with two yeast paralogs that have high sequence similarity and are annotated by the same GO terms)

(7) A general comment: I think the manuscript could be strengthened with more discussion of the motivation for collecting these data and potential questions that can be addressed in future studies. This is a highly unique dataset that I believe will be valuable to our community, but I think this paper will have greater long-term impact if the authors try to highlight some of their motivation for collecting the data and future studies this may enable. The authors do touch on one of the key evolutionary aspects, (e.g. "The extent to which biophysical interactions may occur in the absence of adaptive selection and how such interactions distribute in biological networks remain poorly understood"), but even this discussion could be expanded significantly, including more interpretation of their findings along this direction (i.e. the fact that inter- and intra-species interaction densities are similar). There are many other potential applications waiting to be extracted from these data (e.g. better understanding of host-parasite interactions, discovering binding partners that may be used as drugs, engineering synthetic pathways, etc.), so it seems like a missed opportunity that there isn't more depth in the discussion of motivation/application.

Minor comments:

(1) Figure 5: It is not clear how the genes in the left panels (two of each color) relate to the genes on the right (six of each color), and there is no legend.

(2) The manuscript is readable, but it would benefit from a careful edit: there are misused/missing commas, missing words, and missing letters throughout.

Reviewer #2:

Summary

The authors were investigating the extent to which biophysical interactions among proteins may occur in the absence of natural selection for a long period of time. The authors use a system where two species have diverged for a long period (roughly 1 billion years) such that selection would have played a role in maintaining interactions within each species but not between species. The authors experimentally mapped inter-species protein-protein interactions by performing a Y2H screen between human and yeast proteins. They nicely validated a number of the identified interactions

with LUMIER assays. Their focus was first placed in human proteins that can functionally complement their yeast orthologs. The human rescue genes do tend to share interactions with the yeast genes they functionally replace. They then screened a large portion of the interactome space corresponding to the human and yeast proteins found in a selected reference dataset of each species. They analyzed the resulting interactions for enrichment and various network properties. They find that the inter-species interactome has similar properties to intra-species networks, but they also detect biophysical interactions between proteins with no functional relations, which they call pseudointeractions. The authors suggest that these non-random, non-promiscuous interactions serve as a reservoir for evolutionary innovation.

General remarks

Now that large interactomes exist for multiple model species, there is an increasing interest in studying these interactome under non-standard conditions and test models of how these interactomes may evolve. The current study is an extreme example of this, where all pairs of tested proteins have accumulated over a 3 billions years of evolutionary divergence (1.5 billion years each since the last common ancestor). While this study is not the first to experimentally test protein-protein interactions among different species, it is the first to do so at the interactome level on a large scale and while including proteins that do not have homologs, which would be the major conceptual advance of this study. This study is potentially a major advance in the sense that it allows addressing questions about protein-protein interaction evolution that where not possible without large-scale experimental evidence. The interested audience for this study would be people interested in systems biology, cell biology and evolution. The study is well performed and generally well written. The results are timely and of great interest in a large context. The advance of knowledge could be significant but as described below, more work needs to be done to support the main conclusions the authors would like to reach.

Major points

Overall, the conclusions of the authors are not particularly convincing because it falls a bit beside the general point they make. While the experimental and conceptual designs of this study are impressive, the analysis and discussion accompanying the results could be improved. For example, the title of the paper claims that there is evidence of constrained plasticity in network evolution, but this topic is never directly addressed in the text nor is it explained what "constrained plasticity" could be. The running title "Evolutionary innovation through pseudointeractions" is also misleading because there is no demonstration that pseudointeractions actually lead to evolutionary innovations in this study nor is there any model (mathematical or verbal) that can convincingly show that innovation could take place this way. In fact, the evidence that some of these pseudointeractions are remnant of co-functionality is a stronger point. Such pseudo-interactions that could provide a benefit have been described in another context that not cited in this manuscript (10.1371/journal.pgen.1003836)

At first sight, the fact that the interspecies interactome shows statistical features (Page 7) that are seen in within species interactomes is rather worrisome because it suggests that these characteristics are robust to 3 billions years of divergent evolution, which should be plenty of time to erase much of what natural selection has built. Many of the adaptive explanations that have been put forward to explain these characteristics may therefore not be true and this distribution may simply derive from the intrinsic properties of proteins and not of the specific interactions per se. The authors simply conclude that this means that inter-species interactions are neither random nor promiscuous without further investigation. In the introduction (page 3), the authors mention that they want to test the intrinsic ability of proteins to interact apart from any indirect selective pressure. It would have been interesting to actually see whether it is these intrinsic properties that drive the patterns seen. Because proteins have the same intrinsic properties in the intra and inter species interactome, such properties are very likely to explain the results seen here. For instance, previous studies by the Lehner group and others have shown that disordered proteins are particularly prone at forming protein interactions (<http://www.sciencedirect.com/science/article/pii/S0092867409004541>) and these types of aspects are not considered here. The following questions could therefore have been addressed: Does the size of the protein or level of intrinsic disorder correlate with the number of interactions? Is this the case for both the co-functional and biophysical interactions?

The discussion is very short and would benefit from a larger context. For instance, I could find in the literature few studies that have looked at protein interactions between species and they reach different or similar conclusions as the ones presented here. The authors do not cite any of these previous studies (for instance on page 3), which makes it difficult to examine how generalizable and comparable are their conclusions are (<http://www.pnas.org/content/109/7/E406>, 10.1371/journal.pgen.1003161, 10.1371/journal.pgen.1003836).

« the frequency of inter-species interactions between human and yeast proteins is independent of their levels of sequence conservation » : While this seems to be true for human proteins, there seems to be a correlation for yeast in fig 3A, at least if we understand well the lower panel on the right. It is difficult to see what tests are actually performed.

Some of the figures are particularly poorly described and detailed. Legend of figure 1 for instance lists the panels and does not explain any of the terms used and the schematic. Fig. 1C: legend is incomplete. What are the dotted ellipses? Why are some proteins underlined? Fig. 3: x-axes are not labelled, what is normalized fraction of interaction in each bin?

Introduction, "adaptive selection" is not a common term. People rather say adaptive evolution or positive or negative selection. I am not sure what the authors refer to in this particular case.

On page 5, the authors mention that they identified 284 interactions between zerolog-zerolog pairs. Is this more than expected by chance alone (or less) and what does it mean in terms of observing interactions between proteins that have not evolved together.

In addition, the definition of zerolog should be refined. For instance, there could be proteins that are not shared between humans and *S. cerevisiae* because they evolved in animals after the split with fungi, or because they were recently lost in the lineage leading to *Saccharomyces* or leading to humans. Some may be proteins that recently evolved in the yeast lineage and or that have recently evolved in the human lineage. The predictions for these types of proteins is very different because some have evolved for quite a long of time together if they were in the common ancestor of human and fungi and some may never have evolved together. Interactions among these proteins are thus more surprising.

The term 'pseudointeractions' does not seem appropriate for what is seen here. Pseudo means that these would not be genuine and so it sounds as if they were some sorts of artefacts. However, these interactions take place and whether or not they would result in a functional affect in the cell is a different question. These could thus be called putative non-functional interactions or something along these lines.

This sentence needs to be clarified: "The non-promiscuous nature of the inter-interactome suggests that pseudointeractions may be under the same biochemical or biophysical constraints as functional interactions". Again, I have not seen any analysis in the paper that relates to these biochemical properties that could explain inter-species interactions (see comment above).

I believe some aspects should be discussed in the discussion section in addition to the points made above. For instance, the authors argue that inter-species interactions are not promiscuous. However, there could be promiscuity but this would be manifesting itself as low affinity interactions, which could not be seen here.

1st Revision - authors' response

28 December 2015

We thank the three reviewers for their thoughtful and constructive review of our manuscript currently entitled "An inter-species protein-protein interaction network across vast evolutionary distance".

Reviewer #1:

Summary: This paper presents a new PPI dataset of ~7,000 human proteins screened for physical

interactions with ~4,000 yeast proteins. The analysis characterizes the functional and network properties of the resulting bipartite network. The authors make several conclusions based on this study: they conclude that inter-species network properties are similar to those of intra-species networks in terms of density and degree distribution; orthologs tend to interact with their partner's interactors more than expected by chance; there is limited functional information in the inter-species network, though it is better than random; intra-species degree doesn't correlate to gene properties (e.g. essentiality and pleiotropy) as strongly as intra-species degree does. In general, this is a unique dataset, which will be a valuable resource for several follow-up studies. However, the manuscript could be significantly strengthened with refinements to specific analyses (see details below). Also, more in-depth discussion of the motivation for collecting and potential applications of these data would be helpful.

We sincerely appreciate the reviewer's summary on the strength and weakness of our study and have revised the manuscript accordingly.

Major comments:

(1) Regarding the human-yeast rescue analysis (Fig. 1c): is the fraction of PPI conserved for human orthologs predictive of the ability to rescue? I'm guessing there are several instances of human proteins that didn't rescue their essential yeast orthologs for which the authors also have PPI screen data-- this could be used to contrast the 25% PPI conservation among the "rescuers". A related, but more general question: is the fraction of conserved interactions different for essential vs. non-essential orthologs?

The Reviewer asked two fundamental questions. **First, to what extent cross-species functional complementation may be correlated with the presence of inter-species PPIs.** We cannot address this question using data from Fig 1C (Fig 2A of the revised manuscript) as suggested, since only human proteins that have been reported to complement their yeast orthologs in the literature were screened for interacting yeast proteins in that experiment. We can, however, address this question using our systematic human-yeast inter-interactome, YHII-1.

The lack of a report of any given human protein complementing its orthologs in yeast does not necessarily indicate its lack of rescuing ability. To identify human proteins that cannot complement their yeast orthologs, we considered a recent dataset (Kachroo et al, 2015), in which human proteins were systematically tested for their ability to complement their corresponding orthologs in yeast. We asked if there is an enrichment of human proteins that complement their yeast homologs in the systematic inter-interactome YHII-1.

A total of 294 of the 424 human proteins tested in Kachroo et al. are in the search space for mapping YHII-1. Approximately 5% (15) of these human proteins were found in YHII-1. The small overlap is expected since we did only a single pass Yeast Two-hybrid (Y2H) screen in a single configuration (AD-X_{human} and DB-Y_{yeast}), which is far from reaching screening saturation. Nevertheless, we found a significant enrichment of human proteins in YHII-1 that can complement their yeast orthologs (Odds ratio 3.8. *P*-value = 0.03, Fisher's Exact test). This enrichment is consistent with our conclusions that inter-species interactions significantly correspond to conserved protein binding sites likely underlying cross-species functional complementation.

Among the overlapping 15 proteins, three human-yeast orthologs shared yeast interactors. In all three cases yeast proteins are replaceable by their corresponding human orthologs. For the other nine cases of functional complementation, we have not yet found shared interactors between the human-yeast orthologs. Again, this could be due to the limited coverage of our inter-interactome, YHII-1.

Second, the reviewer suggested a possibility that the levels of conserved inter-species interactions are related to essential gene functions. We addressed this comment by testing the following two predictions.

Frist, would essential genes have more inter-species PPIs than non-essential ones? We compared the average degree between yeast essential genes and non-essential genes in YHII-1 (essential genes are those that make haploid yeast inviable when deleted). When we consider the whole yeast proteome search space of YHII-1, which includes all tested proteins with zero degrees:

Mean number of interactors for essential proteins: 0.77

Mean number of interactors for non-essential proteins: 0.35
 Mann–Whitney U test P -value: 1.77×10^{-11}

When we tested this prediction by considering only yeast proteins that are in the inter-interactome YHII-1:

Mean number of interactors for essential yeast proteins: 3.3
 Mean number of interactors for non-essential yeast proteins: 2.6
 Mann–Whitney U test P -value: 0.009

A second prediction that we tested is whether essential genes tend to have more conserved inter-species interactions. We considered inter-species interactors shared by human-yeast homologous protein pairs (Table EV3) as those that are conserved. Given the incompleteness of intra-species interaction networks, such classification cannot possibly uncover all conserved inter-species interactions. Nevertheless, we found that essential yeast proteins tend to have more of such conserved inter-species interactions than non-essential yeast proteins (Odds ratio 1.8, P -value = 0.001 by Fisher's Exact Test). These findings support that essential proteins tend to have more conserved inter-species interactions.

Results from all three new analyses support that evolutionarily conserved gene function as well as gene essentiality correlate with the presence of inter-species interactions.

We incorporated these results in the revised manuscript (Page 7, 2nd paragraph).

(2) Regarding the results on interaction density: it's surprising (to me, at least) that the density was very similar in both the intra- and inter-species networks. I would have expected either selection for or against interactions in real networks, at least for some proteins. Along these lines, can the authors examine whether there are specific domains or domain-domain pairs that are enriched or depleted in the inter-species network relative to the intra-species network? If so, this could reflect selection for/against interactions with certain domains. This sort of result would deepen this section of the paper. A related question: I may have missed it, but was there a difference in interaction density between conserved-conserved, zerolog-conserved, or zerolog-zerolog pairs? (I guess this analysis is part of Fig. 3A, but the statistics don't seem to be summarized anywhere-see more specific comments on 3A below)

The reviewer made **important suggestions with respect to how we can utilize the systematically mapped inter-species interactome to understand evolutionary selection on biological networks.**

First, following the reviewer's suggestion, we measured the densities of interactions in the subspaces of human-yeast protein pairs that are both conserved, either conserved, or neither conserved between human and yeast. We compared the densities of inter-species interactions to those of the two intra-species parent networks, human HI-1 and yeast YI-1. To differentiate protein pairs that may have never evolved together, we examined density of inter-species interactions amongst lineage-specific human and yeast proteins. We used the HomoloGene Database (<http://www.ncbi.nlm.nih.gov/homologene>) to identify human and yeast proteins that have only metazoan or fungal homologs, respectively. We found that in all subspaces, the densities of inter-species interaction were comparable to intra-species interactions (Fig 4A) even among proteins without any protein domains that are conserved between human and yeast (Fig 4B - the revised version of Fig 3A of our previous submission).

We incorporated results from these analyses (from 3rd paragraph on Page 7 to 1st paragraph on Page 8).

Second, the reviewer kindly suggested that **we could use the human-yeast inter-species interaction network to identify specific interaction domains under selection within species.** To this end, we carried out the following analysis identifying protein domains that have differential interaction propensity across proteomes (Fig 4C). For each yeast or human protein that had their respective DB or AD configuration-specific degree above zero in both inter- and intra- species networks, we calculated the ratio between their configuration-specific degree in the inter-interactome and in the intra-species network ($k_{\text{inter}}/k_{\text{intra}}$). To control for different interaction coverage of the inter- and intra- species networks, we normalized the ratio ($k_{\text{inter}}/k_{\text{intra}}$) by the

average degrees of the inter-interactome and the intra-species network ($(k_{\text{inter}}/k_{\text{intra}})/(\text{Average } k_{\text{inter}}/\text{Average } k_{\text{intra}})$). We ordered the yeast DB-X baits and human AD-Y preys based on their normalized Inter-/Intra- degree ratios. The specific distributions of the normalized Inter-/Intra-degree ratios for sets of proteins with and without particular domains were compared using a Mann-Whitney U test. For empirical statistical comparisons, we created 10,000 randomizations using the following strategy. We preserved the network structures of the inter-interactome and the intra-species yeast and human networks. Protein nodes were binned by the sum of their normalized degree in the inter-interactome and intra-species networks ($k_{\text{total}}=(k_{\text{inter}}/\text{Average } k_{\text{inter}})+(k_{\text{intra}}/\text{Average } k_{\text{intra}})$). Bins included proteins with k_{total} of 1, 2, 3-4, 5-8, 8-16, 17-32, 33-64, 65-128 and 129-256. We permuted the domain annotation vectors amongst proteins in the same bin. This strategy was designed to preserve the general domain annotations of the proteins in the three networks, the correlation between k_{inter} and k_{intra} (Fig 5B), the potential correlation between specific domains and k_{total} , and eliminated the excess $k_{\text{inter}}/k_{\text{intra}}$ that some domains may have. Empirical P -values were obtained by comparing the P -values by the Mann-Whitney U test for each domain in the observed case and to that of the 10,000 randomized situations. Homodimers in the YI-1 and HI-1 network were removed from this analysis for fair comparison to the inter-interactome.

We found three domains that tend to be present in proteins with significantly greater propensity to form inter-species interactions than intra-species interactions (Fig 4C). Among them, the WD40 domain is well known to mediate protein interactions through recognition of diverse short peptides and linear motifs. Linear motifs can arise *de novo* more readily than complex binding interfaces in protein domain. The reduced propensity of proteins with WD40 domain to form intra-species interactions as compared to inter-species interactions suggests selection against the inadvertent binding to linear motifs of these domains within-species, enhancing their binding specificities.

We incorporated results from this analysis in the revised manuscript (2nd paragraph on Page 8).

Third, we also tested **if there would be other biochemical features of protein that can differentiate propensity to form inter-species versus intra-species interactions**. Specifically, we tested Pearson's correlation between the inter-species or intra-species degrees of proteins and several of biochemical features, including contents of intrinsically disordered sequences, Pfam domain coverage, and ORF size.

We predicted disordered residues for each human or yeast ORF using a local installation of the Disopred2 program (Ward et al, 2004). Using a local installation of InterProScan (Zdobnov & Apweiler, 2001), we assigned Pfam domains for each ORF. For human proteins, inter-species degrees correlated positively with the disorder content or proteins but negatively with the coverage of Pfam domains. We did not observe correlation for yeast proteins or between any of the biochemical features of proteins and intra-species degrees.

Biochemical features of proteins	Inter-species degree		Intra-species degree	
	P -value	Correlation	P -value	Correlation
Disorder Content of human AD-X	0.00020	0.17	0.15	0.066
Pfam Domain Coverage of human AD-X	0.016	-0.11	0.40	0.039
ORF Length of human AD-X	0.38	0.041	0.063	-0.086
Disorder Content of yeast DB-X	0.81	0.010	0.83	-0.0091
PFAM Domain Coverage of yeast DB-X	0.27	0.047	0.034	0.089
ORF Length of yeast DB-X	0.93	-0.0035	0.95	-0.0027

These results are consistent with the notion that intrinsically disordered regions of proteins confer conformational flexibility to binding partners (Dunker et al, 2005) and are prone to form promiscuous molecular interactions through mass-action effect (Vavouri et al, 2009). Disordered regions of proteins likely provide an increased tendency for adventitious inter-species interactions. The anti-correlation that we observed between inter-species interactions and the Pfam domain coverage of proteins is consistent with our observation on disordered sequence.

We incorporated results from this analysis in the revised manuscript (1st paragraph on Page 9).

Together, these observations support the idea that similar interaction densities of the inter-interactome and its intra-species parent networks are result of opposing evolutionary forces: i) relative frequent emergence of biophysical interactions uncoupled from pre-existing functional constraints; ii) evolutionary selection that preserves functional interactions and removes deleterious interactions in intra-species networks.

We incorporated results from these analyses (3rd paragraph on Page 7 to 2nd paragraph on Page 9) and a discussion on all of these findings (2nd paragraph on Page 13 to 1st paragraph on Page 14).

(3) The lack of correlation between inter-species degree and essentiality or pleiotropy is one of the more interesting results in the paper. However, I'd suggest laying out the statistics supporting this finding more clearly. First, why is the range of degree (x-axis) in the left panel of Fig. 3C twice as big as the right panel? Does this have to do with how many data are available in each bin? The caption doesn't have any info that might explain this. Also, I find correlations computed on binned, cumulative data misleading. For example, for the essentiality result, I'd suggest just doing a test on the degree distributions of the essential vs. non-essential genes in the two networks (e.g. a Wilcoxon rank-sum test would be appropriate for this). For the "number of phenotypes" result, I would just suggest computing a correlation (either Pearson or Spearman) on the actual degree and the number of phenotypes for each gene (not the cumulative data that's plotted). If correlations computed using this more standard approach are consistent with the current findings, this would strengthen my confidence in this result.

We agree with the reviewer and first examined the degree distributions of the essential versus non-essential genes for intra-species yeast network YI-1:

Mean number of interactors for essential proteins: 2.6

Mean number of interactors for non-essential proteins: 1.9

Mann–Whitney *U* test *P*-value: 0.13

Given that essential genes do not have significantly higher degrees in the intra-species YI-1 network, we have removed this result in the revised manuscript.

Second, we tested the correlation between degree and pleiotropy in yeast YI-1 and inter-interactome YHII-1. We found that the number of phenotypes for yeast proteins is correlated with number of interactors in the intra-species yeast YI-1 network (Pearson's product-moment correlation: 0.22, *P*-value = 0.016). In contrast, the number of phenotypes for yeast proteins is not correlated with number of human interactors in the inter-species YHII-1 network (Pearson's product-moment correlation *P*-value = 0.41)

We revised the manuscript based on the new results (from 4th paragraph on Page 9 to 1st paragraph on Page 10).

(4) Related to the results presented in Fig. 2C: the homology-dependent overlap is a nice result. A related suggestion: can the authors check whether the overall conservation rate of a given protein's interaction neighbors is a predictive feature of the inter-species interaction frequency? For example, for orthologs gH and gY (for human and yeast), if 80% of gY's intra-species interactions are conserved in the human proteome, are there more likely inter-species interactions than if only 20% of gY's interactors are conserved in human? The answer to this would be a useful way of addressing the authors' query about the "extent to which inter-species interactions arise from evolutionarily conserved protein-binding mechanisms" since it is the intra-species interactions that would cause the conservation pressure.

The reviewer suggested an interesting hypothesis that can be tested. **Would the extent to which intra-species interactors of a give protein X are conserved between human and yeast determines the frequency of the detected inter-species interactions of X?**

Following the suggestion, we computed Pearson's correlation between the inter-species degrees of yeast or human proteins and the fractions of their intra-species interactors that are conserved between human and yeast. We computed these correlations in both the full search space as well as in the YHII-1 network space by removing proteins with inter-species degree zero. As a control, we also computed the correlations between the intra-species degrees of yeast or human proteins and the

fractions of their intra-species interactors that are conserved between human and yeast.

Pearson's correlation (P -value)	Fraction of intra-species interactors that are conserved between human and yeast, and	
	Inter-species degree	Intra-species degree
All yeast proteins in the search space	0.16 (0*)	0.33 (0*)
All human proteins in the search space	0.21 (0*)	0.29 (0*)
Yeast proteins in the YHII-1 network	0.10 (0.014)	0.21 (8.2×10^{-7})
Human proteins in the YHII-1 network	0.10 (0.027)	0.23 (2.5×10^{-7})

**P*-value of 0 indicates those that are extremely small and close to 0.

Both the inter- and intra-species degrees of yeast or human proteins are correlated with the fractions of their intra-species interactors that are conserved between human and yeast. This is consistent with our observation that inter- and intra-species degrees are correlated (Fig 5B). We could conclude that both inter- and intra-species interactions arise more frequently from evolutionarily conserved protein-binding sites. However, we have the following three concerns:

- (i) Our inter- and intra-species network maps are still far from reaching complete coverage. Could any experimental bias interfere with our interpretation of these correlations?
- (ii) Does more conserved protein binding sites necessarily lead to higher inter-species degrees? Would different binding sites have different propensity in forming interactions.
- (iii) Does a higher fraction of conserved intra-species interactors necessarily predict more conserved protein binding sites? Co-evolution, for example, preserves interactions while modifying the binding interface and would completely abrogate inter-species interactions among conserved proteins.

Given the potential complications listed above, we chose not to include these results in our revised manuscript. We remain enthusiastically interested in further testing this hypothesis after we obtain more data in both inter and intra-species network mapping, when we might be able to better control effects of co-evolution and degree bias between different proteins. These results, although not included, should not negatively impact our conclusion that inter-species interactions significantly corresponds to conserved protein-binding mechanisms.

(5) Comment on Figure 3A: It is somewhat difficult to understand the setup of this figure. I understand that the blue and yellow bar charts are functioning as x-axis labels for the adjacent histograms, but this could be labeled or stated in the legend. However, it isn't clear what is being measured along this x-axis. The main text ("the frequency of inter-species interactions between human and yeast proteins is independent of their levels of sequence conservation") is vague but seems to refer to conservation between sequences of orthologous pairs. However, the figure caption ("proteins are arranged and binned according to the fraction of their sequences found within protein domains present in both yeast and human proteomes") seems to say that a species-level evolutionarily conserved domain, not necessarily any ortholog, is enough to constitute conservation. Which is correct? And could the authors clarify in both locations? What do the dashed lines represent? Also, I'm not sure what's actually being plotted in the two plots referred to as histograms-I would expect something to sum to one here (either in each bin or across all bins, but that doesn't seem to be true). I think this figure could be simplified and all axes/legend labeled more clearly. It would be interesting to know if there is a difference between degrees of zerologs and orthologs, and it's hard to tell if the authors address this.

We appreciate the suggestions by the reviewer and simplified this figure panel and compared interaction densities among conserved and non-conserved proteins.

We removed the histograms, which appeared to be confusing. We measured the densities of interactions in the subspaces of proteins with or without protein domains conserved between human and yeast. We compared the densities of interactions in the inter-interactome and the two intra-species parent networks, human HI-1 and yeast YI-1.

We also measured the densities of interactions in the subspaces of human-yeast protein pairs that are

both conserved (having homologs in the opposing species), either conserved, or neither conserved between human and yeast. To differentiate protein pairs that may have never evolved together, we examined density of inter-species interactions amongst lineage-specific human and yeast proteins. To this end, we considered the HomoloGene Database (<http://www.ncbi.nlm.nih.gov/homologene>) to identify human and yeast proteins that have only metazoan or fungal homologs, respectively.

In all subspaces, the densities of inter-species interaction were comparable to intra-species interactions (Fig 4A and 4B).

We incorporated results from these analyses (from 3rd paragraph on Page 7 to 1st paragraph on Page 8) and a discussion on these findings (2nd paragraph on Page 13 to 1st paragraph on Page 14).

(6) Regarding the findings in this section: "Correspondence between the inter-interactome and intra-species functional networks", can the authors explain in more detail how they calculated the y-axis on all of these plots in Methods/figure legends (Fig. 4A, EV5). The fold enrichment they are plotting is quite sensitive to the background gene set, which I'm guessing varies between their inter-species and intra-species networks. Also, I noticed that two examples contain yeast duplicate genes (MCM3/MCM4 and SAM1/SAM2). In how many cases does "co-functionality" co-occur with sequence/structural similarity of the proteins? Could intra-species duplication be a confounding factor in the "remnants of co-functionality" result? (e.g. a human protein interacts with two yeast paralogs that have high sequence similarity and are annotated by the same GO terms)

The reviewer expressed concern on the observed remnants of co-functionality in the inter-interactome, questioning whether the functional overlap between inter-species interacting proteins might be simply due to human proteins interacting with yeast paralogs with shared function. We agree that this is a valid concern. To control for the effects of gene duplication, we measured enrichment of GO annotations after removing paralogs from the yeast YI-1 and inter-interactome YHII-1 datasets.

We identified yeast paralogs using the InParanoid (<http://inparanoid.sbc.su.se/cgi-bin/index.cgi>) and the HomoloGene (<http://www.ncbi.nlm.nih.gov/homologene>) databases. If two yeast genes are in the same ortholog group or cluster in either database, they are considered to be a paralog pair. We measured enrichments of GO annotation at four GO specificity cutoffs (5, 20, 100 and 500). Our results (Fig EV1) show that remnants of co-functionality remain significant after removing paralogs.

We incorporated results from these analyses in the revised manuscript (3rd paragraph on Page 10).

(7) A general comment: I think the manuscript could be strengthened with more discussion of the motivation for collecting these data and potential questions that can be addressed in future studies. This is a highly unique dataset that I believe will be valuable to our community, but I think this paper will have greater long-term impact if the authors try to highlight some of their motivation for collecting the data and future studies this may enable. The authors do touch on one of the key evolutionary aspects, (e.g. "The extent to which biophysical interactions may occur in the absence of adaptive selection and how such interactions distribute in biological networks remain poorly understood"), but even this discussion could be expanded significantly, including more interpretation of their findings along this direction (i.e. the fact that inter- and intra-species interaction densities are similar). There are many other potential applications waiting to be extracted from these data (e.g. better understanding of host-parasite interactions, discovering binding partners that may be used as drugs, engineering synthetic pathways, etc.), so it seems like a missed opportunity that there isn't more depth in the discussion of motivation/application.

We appreciate the reviewer's insightful suggestions, and have expanded the discussion (from 3rd paragraph on Page 12 to 2nd paragraph on Page 15) on the utility of the dataset in the following two directions:

A: Protein and network evolution

- emergence of new interaction prior to pre-existing functional relationships
- selection for functional interactions and against deleterious interactions in biological networks
- topological network properties may not be strictly adaptive

B: Gene function

- inference of conservation of gene functions from one-species to the other
- identify the underlying molecular basis of cross-species functional complementation
- inference of new functions emerging from ancestral protein binding capabilities
-

Minor comments:

(1) Figure 5: It is not clear how the genes in the left panels (two of each color) relate to the genes on the right (six of each color), and there is no legend.

We appreciate the reviewer pointing out the ambiguity of Figure 5 and have removed this figure from the revised manuscript.

(2) The manuscript is readable, but it would benefit from a careful edit: there are misused/missing commas, missing words, and missing letters throughout.

We appreciate the reviewer's criticisms have edited the manuscript to improve readability.

Reviewer #2:

Summary: The authors were investigating the extent to which biophysical interactions among proteins may occur in the absence of natural selection for a long period of time. The authors use a system where two species have diverged for a long period (roughly 1 billion years) such that selection would have played a role in maintaining interactions within each species but not between species. The authors experimentally mapped inter-species protein-protein interactions by performing a Y2H screen between human and yeast proteins. They nicely validated a number of the identified interactions with LUMIER assays. Their focus was first placed in human proteins that can functionally complement their yeast orthologs. The human rescue genes do tend to share interactions with the yeast genes they functionally replace. They then screened a large portion of the interactome space corresponding to the human and yeast proteins found in a selected reference dataset of each species. They analyzed the resulting interactions for enrichment and various network properties. They find that the inter-species interactome has similar properties to intra-species networks, but they also detect biophysical interactions between proteins with no functional relations, which they call pseudointeractions. The authors suggest that these non-random, non-promiscuous interactions serve as a reservoir for evolutionary innovation.

General remarks: Now that large interactomes exist for multiple model species, there is an increasing interest in studying these interactome under non-standard conditions and test models of how these interactomes may evolve. The current study is an extreme example of this, where all pairs of tested proteins have accumulated over a 3 billions years of evolutionary divergence (1.5 billion years each since the last common ancestor). While this study is not the first to experimentally test protein-protein interactions among different species, it is the first to do so at the interactome level on a large scale and while including proteins that do not have homologs, which would be the major conceptual advance of this study. This study is potentially a major advance in the sense that it allows addressing questions about protein-protein interaction evolution that where not possible without large-scale experimental evidence. The interested audience for this study would be people interested in systems biology, cell biology and evolution. The study is well performed and generally well written. The results are timely and of great interest in a large context. The advance of knowledge could be significant but as described below, more work needs to be done to support the main conclusions the authors would like to reach.

We sincerely appreciate the reviewer's summary on the strength and weakness of our study and have revised the manuscript accordingly.

Major points

(1) Overall, the conclusions of the authors are not particularly convincing because it falls a bit beside the general point they make. While the experimental and conceptual designs of this study are impressive, the analysis and discussion accompanying the results could be improved. For example, the title of the paper claims that there is evidence of constrained plasticity in network evolution, but this topic is never directly addressed in the text nor is it explained what "constrained plasticity" could be. The running title "Evolutionary innovation through pseudointeractions" is also misleading because there is no demonstration that pseudointeractions actually lead to evolutionary innovations

in this study nor is there any model (mathematical or verbal) that can convincingly show that innovation could take place this way. In fact, the evidence that some of these pseudointeractions are remnant of co-functionality is a stronger point. Such pseudo-interactions that could provide a benefit have been described in another context that not cited in this manuscript (10.1371/journal.pgen.1003836)

The Reviewer challenged how we phrased the main conclusions, the title of the manuscript and the focus on pseudointeractions.

First, following suggestion by the reviewer, we discussed our work in the context of early studies on inter-species interactions. Inter-species protein-protein interactions have been mapped (Diss et al, 2013) to study pathogen-host interactions (Calderwood et al, 2007; Jager et al, 2012; Mukhtar et al, 2011; Pichlmair et al, 2012; Rozenblatt-Rosen et al, 2012; Selbach et al, 2009), to identify evolutionary modifications of protein binding specificity (Das et al, 2013; Zamir et al, 2012; Zarrinpar et al, 2003), and to characterize chimeric protein complexes in hybrid cells of closely related species (Leducq et al, 2012; Piatkowska et al, 2013). The human-yeast inter-interactome presented here is unique in that it is the first inter-species interactome mapped between two evolutionarily distant proteomes. The systematic mapping strategy, the coverage at the proteome-scale, the inclusion of proteins that do not have homologs in the opposing species, the well-controlled experimental conditions matching the inter-interactome to the two intra-species parent networks has allowed us to analyze the commonalities and differences between the inter- and intra-species networks, gaining new insights in the evolution of biological networks and inference of ancestral gene function.

Second, we agree with the reviewer that the title and running title of our first submission were misleading. We revised the title of the manuscript as “An inter-species protein-protein interaction network across vast evolutionary distance”, and the running title as “A human-yeast inter-species interactome”.

Third, we expanded the discussion to cover the utility of the dataset in the following two directions:

A: Protein and network evolution

- emergence of new interaction prior to pre-existing functional relationships
- selection for functional interactions and against deleterious interactions in biological networks
- topological network properties may not be strictly adaptive

B: Gene function

- inference of conservation of gene functions from one-species to the other
- identify the underlying molecular basis of cross-species functional complementation
- inference of new functions emerging from ancestral protein binding capabilities

Finally, we think that non-functional interactions serving as a reservoir to evolve biologically relevant functional interactions is an important point. This is based on our observations of non-conserved proteins interacting with distantly separated homologous protein pairs (Table EV3), forming functionally meaningful inter-species interactions and function enriched network communities in the inter-interactome (Fig 6 and Fig 7). These findings are consistent with conserved proteins acquiring new functions with species-specific new proteins through conserved binding sites (Zeke et al, 2015).

To account for how evolution moves towards complexity and diversity, various models have been put forward, which focus on changes in genes or their regulations (Carroll, 2008; De Robertis, 2008). The appearance of new interactions in proteins has also been demonstrated to involve modifications of pre-existing binding interfaces (Bridgham et al, 2006; Ernst et al, 2009), or formation of new interfaces (Fernandez & Lynch, 2011). Countering evolutionary changes, proteins may need to retain specific ancestral properties due to pre-existing functional constraints. Exploitation of ancestral binding sites for new functions is consistent with gene co-option in evolution (True & Carroll, 2002), when natural selection finds new functions for existing genes. Species-specific new interactions at conserved binding sites may connect ancestral cellular machineries to species-specific functional modules in complex biological systems.

The revised discussion is from 3rd paragraph on Page 12 to 2nd paragraph on Page 15.

(2) At first sight, the fact that the interspecies interactome shows statistical features (Page 7) that are seen in within species interactomes is rather worrisome because it suggests that these characteristics are robust to 3 billions years of divergent evolution, which should be plenty of time to erase much of what natural selection has built. Many of the adaptive explanations that have been put forward to explain these characteristics may therefore not be true and this distribution may simply derive from the intrinsic properties of proteins and not of the specific interactions per se. The authors simply conclude that this means that inter-species interactions are neither random nor promiscuous without further investigation. In the introduction (page 3), the authors mention that they want to test the intrinsic ability of proteins to interact apart from any indirect selective pressure. It would have been interesting to actually see whether it is these intrinsic properties that drive the patterns seen. Because proteins have the same intrinsic properties in the intra and inter species interactome, such properties are very likely to explain the results seen here. For instance, previous studies by the Lehner group and others have shown that disordered proteins are particularly prone at forming protein interactions (<http://www.sciencedirect.com/science/article/pii/S0092867409004541>) and these types of aspects are not considered here. The following questions could therefore have been addressed: Does the size of the protein or level of intrinsic disorder correlate with the number of interactions? Is this the case for both the co-functional and biophysical interactions?

First, we thank the reviewer's comments and have discussed the implication of the observed similarity in network properties of the human-yeast inter-interactome and intra-species networks. We suggest that global topological features of biological networks may not be strictly adaptive. However, the lack of correlation between degree and pleiotropy of yeast proteins in the inter-interactome suggests that biological functions could still influence preservation of certain topological features in interaction networks.

The revised discussion of this point is from 3rd paragraph on Page 9 to 1st paragraph on Page 10 as well as the 2nd paragraph on Page 14.

More importantly, the reviewer challenged our conclusion that inter-species interactions might depend on the same biochemical features of proteins. Given that inter-species interactions are uncoupled from direct evolutionary selection, the reviewer wondered **if there would be detectable difference in the relative contribution of interactions from proteins with distinct biochemical features between the human-yeast inter-interactome and intra-species networks.**

Following the suggestions, we tested Pearson's correlation between the inter-species or intra-species degrees of proteins and several of biochemical features, including contents of intrinsically disordered sequences, Pfam domain coverage, and ORF size.

Biochemical features of proteins	Inter-species degree		Intra-species degree	
	P-value	Correlation	P-value	Correlation
Disorder Content of human AD-X	0.00020	0.17	0.15	0.066
Pfam Domain Coverage of human AD-X	0.016	-0.11	0.40	0.039
ORF Length of human AD-X	0.38	0.041	0.063	-0.086
Disorder Content of yeast DB-X	0.81	0.010	0.83	-0.0091
PFAM Domain Coverage of yeast DB-X	0.27	0.047	0.034	0.089
ORF Length of yeast DB-X	0.93	-0.0035	0.95	-0.0027

As the reviewer rightfully pointed out intrinsically disordered regions of proteins are known to confer conformational flexibility to binding partners (Dunker et al, 2005) and are prone to form promiscuous molecular interactions through mass-action effect (Vavouri et al, 2009). For human proteins, inter-species degrees correlated positively with the disorder content or proteins but negatively with the coverage of Pfam domains. We did not observe correlation for yeast proteins or

between any of the three biochemical features of proteins and intra-species degrees. These observations are consistent with disordered regions of human proteins providing an increased tendency for adventitious inter-species interactions.

We incorporated results from this analysis in the revised manuscript (1st paragraph on Page 9).

In addition to disorder content in proteins, we also tested **if there are specific protein domains that might have increased tendency to form inter-species interactions relative to intra-species interactions.**

We carried out the following analysis identifying protein domains that have differential interaction propensity across proteomes (Fig 4C). For each yeast or human protein that had their respective DB or AD configuration-specific degree above zero in both inter- and intra- species networks, we calculated the ratio between their configuration-specific degree in the inter-interactome and in the intra-species network ($k_{\text{inter}}/k_{\text{intra}}$). To control for different interaction coverage of the inter- and intra-species networks, we normalized the ratio ($k_{\text{inter}}/k_{\text{intra}}$) by the average degrees of the inter-interactome and the intra-species network ($(k_{\text{inter}}/k_{\text{intra}})/(\text{Average } k_{\text{inter}}/\text{Average } k_{\text{intra}})$). We ordered the yeast DB-X baits and human AD-Y preys based on their normalized Inter-/Intra- degree ratios. The specific distributions of the normalized Inter-/Intra- degree ratios for sets of proteins with and without particular domains were compared using a Mann-Whitney U test. For empirical statistical comparisons, we created 10,000 randomizations using the following strategy. We preserved the network structures of the inter-interactome and the intra-species yeast and human networks. Protein nodes were binned by the sum of their normalized degree in the inter-interactome and intra-species networks ($k_{\text{total}}=(k_{\text{inter}}/\text{Average } k_{\text{inter}})+(k_{\text{intra}}/\text{Average } k_{\text{intra}})$). Bins included proteins with k_{total} of 1, 2, 3-4, 5-8, 8-16, 17-32, 33-64, 65-128 and 129-256. We permuted the domain annotation vectors amongst proteins in the same bin. This strategy was designed to preserve the general domain annotations of the proteins in the three networks, the correlation between k_{inter} and k_{intra} (Fig 5B), the potential correlation between specific domains and k_{total} , and eliminated the excess $k_{\text{inter}}/k_{\text{intra}}$ that some domains may have. Empirical P values were obtained by comparing the P values by the Mann-Whitney U test for each domain in the observed case and to that of the 10,000 randomized situations. Homodimers in the YI-1 and HI-1 network were removed from this analysis for fair comparison to the inter-interactome.

We found three domains that tend to be present in proteins with significantly greater propensity to form inter-species interactions than intra-species interactions (Fig 4C). Among them, the WD40 domain is well known to mediate protein interactions through recognition of diverse short peptides and linear motifs. Linear motifs can arise *de novo* more readily than complex binding interfaces in protein domain. The reduced propensity of proteins with WD40 domain to form intra-species interactions as compared to inter-species interactions suggests selection against the inadvertent binding to linear motifs of these domains within-species, enhancing their binding specificities.

We incorporated results from this analysis in the revised manuscript (2nd paragraph on Page 8).

Together, these observations support the idea that similar interaction densities of the inter-interactome and its intra-species parent networks are result of opposing evolutionary forces: i) relative frequent emergence of biophysical interactions uncoupled from pre-existing functional constraints; ii) evolutionary selection that preserves functional interactions and removes deleterious interactions in intra-species networks.

The discussion for all of the results above is included from 4th paragraph on Page 13 to 1st paragraph on Page 14.

(3) The discussion is very short and would benefit from a larger context. For instance, I could find in the literature few studies that have looked at protein interactions between species and they reach different or similar conclusions as the ones presented here. The authors do not cite any of these previous studies (for instance on page 3), which makes it difficult to examine how generalizable and comparable are their conclusions are (<http://www.pnas.org/content/109/7/E406>, [10.1371/journal.pgen.1003161](http://www.pnas.org/content/109/7/E406), [10.1371/journal.pgen.1003836](http://www.pnas.org/content/109/7/E406)).

Following suggestion by the reviewer, we discussed our work in the context of early studies on

inter-species interactions. Inter-species protein-protein interactions have been mapped (Diss et al, 2013) to study pathogen-host interactions (Calderwood et al, 2007; Jager et al, 2012; Mukhtar et al, 2011; Pichlmair et al, 2012; Rozenblatt-Rosen et al, 2012; Selbach et al, 2009), to identify evolutionary modifications of protein binding specificity (Das et al, 2013; Zamir et al, 2012; Zarrinpar et al, 2003), and to characterize chimeric protein complexes in hybrid cells of closely related species (Leducq et al, 2012; Piatkowska et al, 2013). The human-yeast inter-interactome presented here is unique in that it is the first inter-species interactome mapped between two evolutionarily distant proteomes. The systematic mapping strategy, the coverage at the proteome-scale, the inclusion of proteins that do not have homologs in the opposing species, the well-controlled experimental conditions matching the inter-interactome to two intra-species parent networks has allowed us to analyze the commonalities and differences between the inter- and intra-species networks, gaining new insights in the evolution of biological networks and inference of ancestral gene function. We expanded the discussion to cover two major utility of systematic mapping of inter-species network emphasize the utility of the dataset as discussed above.

The revised discussion is from 3rd paragraph on Page 12 to 2nd paragraph on Page 15.

(4) « the frequency of inter-species interactions between human and yeast proteins is independent of their levels of sequence conservation » : While this seems to be true for human proteins, there seems to be a correlation for yeast in fig 3A, at least if we understand well the lower panel on the right. It is difficult to see what tests are actually performed.

We appreciate the suggestions by the reviewer and simplified this figure panel. We removed the histograms, which appeared to be confusing. We measured the densities of interactions in the subspaces of proteins with or without protein domains conserved between human and yeast. We compared the densities of interactions in the inter-interactome and the two intra-species parent networks, human HI-1 and yeast YI-1.

We also measured the densities of interactions in the subspaces of human-yeast protein pairs that are both conserved (having homologs in the opposing species), either conserved, or neither conserved between human and yeast. To differentiate protein pairs that may have never evolved together, we examined density of inter-species interactions amongst lineage-specific human and yeast proteins. To this end, we considered the HomoloGene Database (<http://www.ncbi.nlm.nih.gov/homologene>) to identify human and yeast proteins that have only metazoan or fungal homologs, respectively.

In all subspaces, the densities of inter-species interaction were comparable to intra-species interactions (Fig 4A and 4B).

We incorporated results from these analyses (from 3rd paragraph on Page 7 to 1st paragraph on Page 8) and a discussion on these findings (2nd paragraph on Page 13 to 1st paragraph on Page 14).

(5) Some of the figures are particularly poorly described and detailed. Legend of figure 1 for instance lists the panels and does not explain any of the terms used and the schematic. Fig. 1C: legend is incomplete. What are the dotted ellipses? Why are some proteins underlined? Fig. 3: x-axes are not labeled, what is normalized fraction of interaction in each bin?

We have fixed these issues in the figures and figure legends.

(6) Introduction, "adaptive selection" is not a common term. People rather say adaptive evolution or positive or negative selection. I am nor sure what the authors refer to in this particular case.

We agree with the reviewer and have changed the text to “adaptive evolution”. (1st paragraph on Page 3 and 2nd paragraph on Page 8).

(7) On page 5, the authors mention that they identified 284 interactions between zerolog-zerolog pairs. Is this more than expected by chance alone (or less) and what does it mean in terms of observing interactions between proteins that have not evolved together.

The reviewer asked an important question with respect to **how pairs of proteins that have not evolved together may form biophysical interactions with each other**. We measured the densities

of interactions in the subspaces of human-yeast protein pairs that are both conserved (having homologs in the opposing species), either conserved, or neither conserved between human and yeast. We compared the densities of inter-species interactions to those of the two intra-species parent networks, human HI-1 and yeast YI-1. We found that in all subspaces, the densities of inter-species interaction were comparable to intra-species interactions (Fig 4A) even among proteins without any protein domains that are conserved between human and yeast (Fig 4B - the revised version of Fig 3A of our previous submission).

We incorporated results from these analyses (from 3rd paragraph on Page 7 to 1st paragraph on Page 8), and a discussion on these findings from 4th paragraph on Page 13 to 1st paragraph on Page 14.

(8) In addition, the definition of zerolog should be refined. For instance, there could be proteins that are not shared between humans and S. cerevisiae because they evolved in animals after the split with fungi, or because they were recently lost in the lineage leading to Saccharomyces or leading to humans. Some may be proteins that recently evolved in the yeast lineage and or that have recently evolved in the human lineage. The predictions for these types of proteins is very different because some have evolved for quite a long of time together if they were in the common ancestor of human and fungi and some may never have evolved together. Interactions among these proteins are thus more surprising.

This is related to the reviewer's point (7). Following the reviewer's suggestion, we examined density of inter-species interactions amongst lineage-specific human and yeast proteins. We used the HomoloGene Database (<http://www.ncbi.nlm.nih.gov/homologene>) to identify human and yeast proteins that have only metazoan or fungal homologs, respectively. We found that the densities of inter-species interaction were comparable to intra-species interactions (Fig 4A)

We incorporated results from these analyses (from 3rd paragraph on Page 7 to 1st paragraph on Page 8) and a discussion on these findings (from 3rd paragraph on Page 13 to 1st paragraph on Page 14).

(9)The term 'pseudointeractions' does not seem appropriate for what is seen here. Pseudo means that these would not be genuine and so it sounds as if they were some sorts of artefacts. However, these interactions take place and whether or not they would result in a functional affect in the cell is a different question. These could thus be called putative non-functional interactions or something along these lines.

We agree with the reviewer and changed the text to non-functional interactions. (Abstract as well as in the 3rd and 4th paragraphs on Page 13).

(10) This sentence needs to be clarified: "The non-promiscuous nature of the inter-interactome suggests that pseudointeractions may be under the same biochemical or biophysical constraints as functional interactions". Again, I have not seen any analysis in the paper that relates to these biochemical properties that could explain inter-species interactions (see comment above).

We agree with the reviewer and have examined **how biochemical properties of proteins may influence the frequency of inter- versus intra-species interactions.**

As discussed above, we tested Pearson's correlation between the inter-species or intra-species degrees of proteins and intrinsically disordered sequences. Human proteins with higher disorder content have greater propensities to form inter-species interactions (Pearson's correlation 0.17, P -value = 0.0002). Such a correlation is absent for intra-species interactions in the human HI-1 network (P -value = 0.2).

These observations are consistent with disordered regions of human proteins providing an increased tendency for adventitious inter-species interactions.

We incorporated results from this analysis in the revised manuscript (1st paragraph on Page 9).

In addition to disorder content in proteins, we also tested **if there are specific protein domains that might have increased tendency to form inter-species interactions relative to intra-species interactions.**

We found three domains that tend to be present in proteins with significantly greater propensity to form inter-species interactions than intra-species interactions (Fig 4C). Among them, the WD40 domain is well known to mediate protein interactions through recognition of diverse short peptides and linear motifs. Linear motifs can arise *de novo* more readily than complex binding interfaces in protein domain. The reduced propensity of proteins with WD40 domain to form intra-species interactions as compared to inter-species interactions suggests selection against the inadvertent binding to linear motifs of these domains within-species, enhancing their binding specificities.

We incorporated results from this analysis in the revised manuscript (2nd paragraph on Page 8).

Together, these observations support the idea that similar interaction densities of the inter-interactome and its intra-species parent networks are result of opposing evolutionary forces: i) relative frequent emergence of biophysical interactions uncoupled from pre-existing functional constraints; ii) evolutionary selection that preserves functional interactions and removes deleterious interactions in intra-species networks.

In summary, we agree with the reviewer that distinct biochemical properties of proteins may contribute differently to inter- and intra-species interactions. We therefore have removed the sentence “The non-promiscuous nature of the inter-interactome suggests that pseudointeractions may be under the same biochemical or biophysical constraints as functional interactions” in the revised manuscript.

The discussion on these new results is included from 4th paragraph on Page 13 to 1st paragraph on Page 14.

(11) I believe some aspects should be discussed in the discussion section in addition to the points made above. For instance, the authors argue that inter-species interactions are not promiscuous. However, there could be promiscuity but this would be manifesting itself as low affinity interactions, which could not be seen here.

We thank the reviewer’s comments and agree that inter-species interactions mapped here are robustly detectable biophysical interactions, which are different from promiscuous sticky events occurring at random. Non-functional interactions may exist in different forms. Some may be robustly detectable as human-yeast inter-species interactions. In crowded cellular environment, however, relatively low affinity interactions might occur at much higher frequency (Deeds et al, 2007; Zhang et al, 2008).

The discussion of this point is in the 3rd paragraph on Page 13.

Reviewer #3:

The paper by Vidal, Roth and colleagues describes a study where they created an inter-interactome by analyzing protein-protein interactions via yeast two-hybrid using 7000 human genes X 3000 human genes and identified 1600 high confidence PPIs. They compare the data to a relevant recent paper by Marcotte and colleagues who carried out a study where they used human genes to complement yeast mutations (Figure 1). They also analyze this data in the context of orthogonal data including domain and structural information (Figure 2), other metrics including "betweenness", scale-free, etc. (Figure 3) as well as other information such as GO, genetic interaction data and co-expression (Figure 4). Their last figure offers a framework on how to interpret the work (Figure 5). While clearly a great deal of work and conceptually a neat idea, I feel that there should be more than just a cross-species dataset, a few trends reported by comparing to orthogonal data types and some theories relating to evolution. This paper would be so much more powerful if there was some mechanism extracted from the datasets about the function of specific pathways, complexes and proteins. One common complaint of the two-hybrid approach is that is an artificial system that is too far away from the real biology; this study potentially removes it even further by going across species. I think the authors could (and should) help alleviate this criticism by extracting out some important biology from this data.

We appreciate the reviewer’s comments and have revised our manuscript by highlighting examples of pathways and proteins to better illustrate our conclusion (from the 4th paragraph on Page 11 to the 2nd paragraph on Page 12).

Second, we expanded the discussion to cover the utility of the dataset in the following two directions.

A: Protein and network evolution

- emergence of new interaction prior to pre-existing functional relationships
- selection for functional interactions and against deleterious interactions in biological networks
- topological network properties may not be strictly adaptive

B: Gene function

- inference of conservation of gene functions from one-species to the other
- identify the underlying molecular basis of cross-species functional complementation
- inference of new functions emerging from ancestral protein binding capabilities

The revised discussion from 3rd paragraph on Page 12 to 2nd paragraph on Page 15 is as follows:

“Inter-species protein-protein interactions have been mapped (Diss et al, 2013) to study pathogen-host interactions (Calderwood et al, 2007; Jager et al, 2012; Mukhtar et al, 2011; Pichlmair et al, 2012; Rozenblatt-Rosen et al, 2012; Selbach et al, 2009), to identify evolutionary modifications of protein binding specificity (Das et al, 2013; Zamir et al, 2012; Zarrinpar et al, 2003), and to characterize chimeric protein complexes in hybrid cells of closely related species (Leducq et al, 2012; Piatkowska et al, 2013). The human-yeast inter-interactome presented here is unique in that it is the first inter-species interactome mapped between two evolutionarily distant proteomes. The systematic mapping strategy, the coverage at proteome-scale, the inclusion of proteins that do not have homologs in the opposing species, and the well-controlled experimental conditions matching the inter-interactome to two intra-species parent networks has allowed us to analyze the commonalities and differences between inter- and intra-species networks, gaining insights in the evolution of biological networks and ancestral gene function.

Inter-interactome mapping as a new way to study network evolution

How biological systems evolve toward complexity and diversity is a central question (Carroll, 2001). Studies on the evolution of cellular networks have primarily focused on understanding how interactions vary over evolutionary time periods between phylogenetically conserved proteins (Sharan & Ideker, 2006). Critically unresolved are two fundamental issues: i) how new interactions arise, and ii) how biophysical and functional interactome networks remain coordinated over evolutionary time (Fig 1).

Our findings that human-yeast inter-species interactions uncoupled from direct selective pressure are as prevalent as intra-species interactions (Fig 4) support the existence of “pseudointeractions” (Venkatesan et al, 2009), *i.e.* interactions between proteins with little or no functional relationships. Some non-functional interactions may be as robustly detectable as human-yeast inter-species interactions, while others may be relatively low affinity interactions occurring at much higher frequency (Deeds et al, 2007; Zhang et al, 2008). Such functionally insignificant interactions (Gray et al, 2010; Levy et al, 2009; Venkatesan et al, 2009), if not purged by purifying selection or lost due to genetic drift (Fernandez & Lynch, 2011), may loosen the correspondence between biophysical interaction and functional networks.

Several protein domains exhibit reduced interaction propensity within species relative to between species (Fig 4C). Combined with evidence that non-functional interactions may be more frequent in the inter-interactome than within species networks (Fig 6), this finding supports opposing evolutionary forces on biological networks. Natural selection acts not only to retain functional interactions, as is widely appreciated, but also to remove deleterious interactions (Zarrinpar et al, 2003), strengthening the correspondence between biophysical and functional networks.

The resemblance between the inter-interactome and the two intra-species parent networks (Fig 4-5) argues that, instead of being strictly adaptive, global topological features of biological networks might at least in part reflect intrinsic properties of proteins.

Inter-interactome mapping as a new way to study gene function

Vastly different organisms can be related to one another genetically through the presence of

conserved genes in their genomes, reflecting the common origin of living organisms and evolutionary constraints on gene function. Ancestral function in proteins is frequently inferred through cross-species comparisons. Cross-species genetic complementation experiments (Dolinski & Botstein, 2007; Kachroo et al, 2015) indicate conserved function between orthologs. The molecular basis of such complementation is often left uncharacterized. Comparative interactomic approaches reveal cross-species conserved molecular interactions, but cannot readily identify conserved binding specificity in proteins, as co-evolved interacting protein pairs may alter ancestral binding capabilities (Zamir et al, 2012).

Mapping inter-species interactions between distant proteomes has identified overlapping interactors between distantly related homologous protein pairs (Fig 2A, Fig 3A and Table EV3). Such inter-species molecular complementarities in proteins may correspond to evolutionarily conserved ancestral binding sites and underlie cross-species functional complementation. In addition to mediating conserved interactions between different species, conserved protein-binding sites may mediate species-specific interactions. Proteins that interact with both human-yeast homologs do not necessarily themselves to be conserved between human and yeast (Fig 2A and Table EV3). Non-conserved proteins appear to mediate functionally meaningful inter-species interactions (Fig 6A), and form network communities enriched for conserved or species-specific functions (Fig 6 and Table EV6).

How could non-conserved proteins form inter-species interactions at ancestral binding sites? First, some proteins, due to their structural modularity (Chothia et al, 2003), may undergo gross sequence variations yet still preserve certain ancestral structural features required for specific interactions. Second, species-specific gene loss (Koonin, 2003) may lead to conserved proteins interacting with proteins maintained in one species but lost in the other. Preservation of ancestral binding sites may be due to functional constraints of any protein interactor that bind to the same binding sites (Kim et al, 2006). Third, conserved proteins may acquire new functions with species-specific new proteins through conserved binding sites (Zeke et al, 2015).

To account for how evolution moves towards complexity and diversity various models have been put forward, which focus on changes in genes or their regulations (Carroll, 2008; De Robertis, 2008). The appearance of new interactions in proteins also involves modifications of pre-existing binding interfaces (Bridgham et al, 2006; Ernst et al, 2009), or formation of new interfaces (Fernandez & Lynch, 2011). Countering evolutionary changes, proteins may need to retain specific ancestral properties due to pre-existing functional constraints. Species-specific new interactions at conserved binding sites may connect ancestral cellular machineries to species-specific functional modules in complex biological systems. Exploitation of ancestral binding sites for new functions is consistent with gene co-option in evolution (True & Carroll, 2002), when natural selection finds new functions for existing genes. Biological networks may therefore evolve in the absence of gross changes in protein binding capability. Just as sequence conservation indicates crucial functional constraints on genomes, inter-species interactome mapping may reveal crucial functional constraints on biological networks.”

Finally, we would like to summarize that our manuscript introduces an unprecedented experimental approach to test the extent to which two eukaryotic proteomes separated by a billion years of evolution can still form biophysical interactions. For the first time, our paper can reject the idea of non-functional biophysical interactions being either completely absent or promiscuous in interactome networks. Such interactions appear to be under negative selection within species, serving as a reservoir to evolve novel biologically relevant interactions. These observations open the doors to a new set of developments in evolutionary network biology. Among such developments are those that critically address fundamental research areas missing from our understanding: i) how new interactions arise, and ii) how biophysical and functional interactome networks remain coordinated over evolutionary time. We therefore argue that systematically mapping biophysical interactions between evolutionarily separated proteomes, and subsequent functional characterization of the resulting inter-species interactome network does help gaining biological insights.

Other comments:

1) When the comparing to the Marcotte study, they looked at a subset of these and claimed there was an enrichment of interactions shared between yeast and human homologs when compared to randomized networks. However, the much more important question is: are these enriched when

compared to other genes that cannot be complemented using human orthologs. Can this analysis be done?

The Reviewer asked an important question, which is also raised by reviewer #1. **To what extent cross-species functional complementation may be correlated with the presence of inter-species PPIs?**

A total of 294 of the 424 human proteins tested in Kachroo et al. (Kachroo et al, 2015) are in the search space for mapping YHII-1. Approximately 5% (15) of these human proteins were found in YHII-1. The small overlap is expected since we did only a single pass Yeast Two-hybrid (Y2H) screen in a single configuration (AD- X_{human} and DB- Y_{yeast}), which is far from reaching screening saturation. Nevertheless, there is a significant enrichment of human proteins in YHII-1 that can complement their yeast orthologs (Odds ratio 3.8. P -value = 0.03 by Fisher's exact test). This enrichment supports our conclusions that cross-species PPIs significantly correspond to ancestral protein binding sites preserved in yeast and human proteomes, which likely underlie functional complementation. Among the overlapping 15 proteins, three human-yeast orthologs shared yeast interactors. In all three cases yeast proteins are replaceable by their corresponding human orthologs. For the other nine cases of functional complementation, we have not yet found shared interactors between the human-yeast orthologs. Again, this could be due to the limited coverage of our inter-interactome, YHII-1. These observations support that cross-species conservation of gene function correlate with the presence of inter-species interactions.

We incorporated these results in the revised manuscript (Page 7, 2nd paragraph).

2) A zerolog is hard to prove as you may need other proteins present to see an interaction, or PTMs, etc... I think this is a dangerous term that at the end of the day could end up just relating to a technique rather than having biological significance.

The reviewer might have misunderstood zerolog as proteins for which we did not identify any interaction (degree as zero). Instead, we used "zerolog" in our previously submitted manuscript to describe proteins that are not conserved between human and yeast. Considering potential confusion caused by using this word, we have removed this word in our revised manuscript and specifically describe them as those proteins that are not conserved between yeast to human.

We appreciate the reviewer pointing out possible bias of specific method for mapping interactions, which we certainly agree. In the past, we have systematically measured such bias of different methods (Yu 2008, Venkatesan 2009, Braun 2009). We would like to emphasize, however, here we carefully controlled all our analyses on the human-yeast inter-interactome using the parent intra-species human and yeast networks. The three networks were mapped using the same high-throughput Y2H method, the same gene clones and under similar experimental conditions. Thus, our conclusions should not be due to a technical bias.

3) It would be more intuitive to assess the figures as they would exist in the paper (not split up into different pieces on several pages)

We appreciate the reviewer's suggestion and have generated complete figures including all panels.

4) What is the biological significance of being or not being "scale-free"?

We thank the reviewer's comments and have discussed this point in the revised manuscript. The resemblance between the inter-interactome and the two intra-species parent networks (Fig 4-5) argues that, instead of being strictly adaptive, global topological features of biological networks might at least in part reflect intrinsic properties of proteins.

The revised discussion of this point is from 3rd paragraph on Page 9 to 1st paragraph on Page 10 as well as the 2nd paragraph on Page 14.

Reference

- Bridgham JT, Carroll SM, Thornton JW (2006) Evolution of hormone-receptor complexity by molecular exploitation. *Science* **312**: 97-101
- Calderwood MA, Venkatesan K, Xing L, Chase MR, Vazquez A, Holthaus AM, Ewence AE, Li N, Hirozane-Kishikawa T, Hill DE, Vidal M, Kieff E, Johannsen E (2007) Epstein-Barr virus and virus human protein interaction maps. *Proc Natl Acad Sci U S A* **104**: 7606-7611
- Carroll SB (2001) Chance and necessity: the evolution of morphological complexity and diversity. *Nature* **409**: 1102-1109
- Carroll SB (2008) Evo-devo and an expanding evolutionary synthesis: a genetic theory of morphological evolution. *Cell* **134**: 25-36
- Chothia C, Gough J, Vogel C, Teichmann SA (2003) Evolution of the protein repertoire. *Science* **300**: 1701-1703
- Das J, Vo TV, Wei X, Mellor JC, Tong V, Degatano AG, Wang X, Wang L, Cordero NA, Kruer-Zerhusen N, Matsuyama A, Pleiss JA, Lipkin SM, Yoshida M, Roth FP, Yu H (2013) Cross-species protein interactome mapping reveals species-specific wiring of stress response pathways. *Sci Signal* **6**: ra38
- De Robertis EM (2008) Evo-devo: variations on ancestral themes. *Cell* **132**: 185-195
- Deeds EJ, Ashenberg O, Gerardin J, Shakhnovich EI (2007) Robust protein protein interactions in crowded cellular environments. *Proc Natl Acad Sci U S A* **104**: 14952-14957
- Diss G, Filteau M, Freschi L, Leducq JB, Rochette S, Torres-Quiroz F, Landry CR (2013) Integrative avenues for exploring the dynamics and evolution of protein interaction networks. *Curr Opin Biotechnol* **24**: 775-783
- Dolinski K, Botstein D (2007) Orthology and functional conservation in eukaryotes. *Annu Rev Genet* **41**: 465-507
- Dunker AK, Cortese MS, Romero P, Iakoucheva LM, Uversky VN (2005) Flexible nets. The roles of intrinsic disorder in protein interaction networks. *FEBS J* **272**: 5129-5148
- Ernst A, Sazinsky SL, Hui S, Currell B, Dharsee M, Seshagiri S, Bader GD, Sidhu SS (2009) Rapid evolution of functional complexity in a domain family. *Sci Signal* **2**: ra50
- Fernandez A, Lynch M (2011) Non-adaptive origins of interactome complexity. *Nature* **474**: 502-505
- Gray MW, Lukes J, Archibald JM, Keeling PJ, Doolittle WF (2010) Cell biology. Irremediable complexity? *Science* **330**: 920-921
- Jager S, Cimermanic P, Gulbahce N, Johnson JR, McGovern KE, Clarke SC, Shales M, Mercenne G, Pache L, Li K, Hernandez H, Jang GM, Roth SL, Akiva E, Marlett J, Stephens M, D'Orso I, Fernandes J, Fahey M, Mahon C et al (2012) Global landscape of HIV-human protein complexes. *Nature* **481**: 365-370
- Kachroo AH, Laurent JM, Yellman CM, Meyer AG, Wilke CO, Marcotte EM (2015) Evolution. Systematic humanization of yeast genes reveals conserved functions and genetic modularity. *Science* **348**: 921-925
- Kim PM, Lu LJ, Xia Y, Gerstein MB (2006) Relating three-dimensional structures to protein networks provides evolutionary insights. *Science* **314**: 1938-1941
- Koonin EV (2003) Comparative genomics, minimal gene-sets and the last universal common ancestor. *Nat Rev Microbiol* **1**: 127-136

- Leducq JB, Charron G, Diss G, Gagnon-Arsenault I, Dube AK, Landry CR (2012) Evidence for the robustness of protein complexes to inter-species hybridization. *PLoS Genet* **8**: e1003161
- Levy ED, Landry CR, Michnick SW (2009) How perfect can protein interactomes be? *Sci Signal* **2**: pe11
- Mukhtar MS, Carvunis AR, Dreze M, Epple P, Steinbrenner J, Moore J, Tasan M, Galli M, Hao T, Nishimura MT, Pevzner SJ, Donovan SE, Ghamsari L, Santhanam B, Romero V, Poulin MM, Gebreab F, Gutierrez BJ, Tam S, Monachello D et al (2011) Independently evolved virulence effectors converge onto hubs in a plant immune system network. *Science* **333**: 596-601
- Piatkowska EM, Naseeb S, Knight D, Delneri D (2013) Chimeric protein complexes in hybrid species generate novel phenotypes. *PLoS Genet* **9**: e1003836
- Pichlmair A, Kandasamy K, Alvisi G, Mulhern O, Sacco R, Habjan M, Binder M, Stefanovic A, Eberle CA, Goncalves A, Burckstummer T, Muller AC, Fauster A, Holze C, Lindsten K, Goodbourn S, Kochs G, Weber F, Bartenschlager R, Bowie AG et al (2012) Viral immune modulators perturb the human molecular network by common and unique strategies. *Nature* **487**: 486-490
- Rozenblatt-Rosen O, Deo RC, Padi M, Adelmant G, Calderwood MA, Rolland T, Grace M, Dricot A, Askenazi M, Tavares M, Pevzner SJ, Abderazzaq F, Byrdsong D, Carvunis AR, Chen AA, Cheng J, Correll M, Duarte M, Fan C, Feltkamp MC et al (2012) Interpreting cancer genomes using systematic host network perturbations by tumour virus proteins. *Nature* **487**: 491-495
- Selbach M, Paul FE, Brandt S, Guye P, Daumke O, Backert S, Dehio C, Mann M (2009) Host cell interactome of tyrosine-phosphorylated bacterial proteins. *Cell Host Microbe* **5**: 397-403
- Sharan R, Ideker T (2006) Modeling cellular machinery through biological network comparison. *Nat Biotechnol* **24**: 427-433
- True JR, Carroll SB (2002) Gene co-option in physiological and morphological evolution. *Annu Rev Cell Dev Biol* **18**: 53-80
- Vavouri T, Semple JI, Garcia-Verdugo R, Lehner B (2009) Intrinsic protein disorder and interaction promiscuity are widely associated with dosage sensitivity. *Cell* **138**: 198-208
- Venkatesan K, Rual JF, Vazquez A, Stelzl U, Lemmens I, Hirozane-Kishikawa T, Hao T, Zenkner M, Xin X, Goh KI, Yildirim MA, Simonis N, Heinzmann K, Gebreab F, Sahalie JM, Cevik S, Simon C, de Smet AS, Dann E, Smolyar A et al (2009) An empirical framework for binary interactome mapping. *Nat Methods* **6**: 83-90
- Ward JJ, Sodhi JS, McGuffin LJ, Buxton BF, Jones DT (2004) Prediction and functional analysis of native disorder in proteins from the three kingdoms of life. *J Mol Biol* **337**: 635-645
- Zamir L, Zaretsky M, Fridman Y, Ner-Gaon H, Rubin E, Aharoni A (2012) Tight coevolution of proliferating cell nuclear antigen (PCNA)-partner interaction networks in fungi leads to interspecies network incompatibility. *Proc Natl Acad Sci U S A* **109**: E406-414
- Zarrinpar A, Park SH, Lim WA (2003) Optimization of specificity in a cellular protein interaction network by negative selection. *Nature* **426**: 676-680
- Zdobnov EM, Apweiler R (2001) InterProScan--an integration platform for the signature-recognition methods in InterPro. *Bioinformatics* **17**: 847-848
- Zeke A, Bastys T, Alexa A, Garai A, Meszaros B, Kirsch K, Dosztanyi Z, Kalinina OV, Remenyi A (2015) Systematic discovery of linear binding motifs targeting an ancient protein interaction surface on MAP kinases. *Mol Syst Biol* **11**: 837
- Zhang J, Maslov S, Shakhnovich EI (2008) Constraints imposed by non-functional protein-protein interactions on gene expression and proteome size. *Mol Syst Biol* **4**: 210

Thank you again for submitting your work to Molecular Systems Biology. We are now satisfied with the modifications made and we will be able to accept your paper for publication in Molecular Systems Biology following minor amendments.

REFeree COMMENTS

Reviewer #2:

I have read the responses to my previous comments and they appear satisfactory. The manuscript is greatly improved.

To the editor:

This manuscript would be great material for a News and Views as it touches many aspects of systems biology, including evolution, which is rarely investigated experimentally at the level of protein interaction networks.

Corresponding Author Name:
Journal Submitted to:
Manuscript Number: